# Study on Spark Image Detection for Abrasive Belt Grinding via Transfer Learning with YOLOv8

**DOI:** 10.3390/s25092946

**Published:** 2025-05-07

**Authors:** Jian Huang, Guangpeng Zhang

**Affiliations:** 1School of Mechanical and Precision Instrument Engineering, Xi’an University of Technology, Xi’an 710048, China; 2School of Computer Science, Xijing University, Xi’an 710123, China

**Keywords:** yolov8, deep learning, transfer learning, spark image

## Abstract

Aiming to solve the problems of low precision and poor efficiency caused by relying on manual experience during the manual polishing of blades, a multi-view spark image detection method based on YOLOv8 transfer learning is proposed. A multi-pose spark image dataset including front, side, and 45° angle views is constructed, and the cross-view detection task is achieved for the first time. The generalization ability of the model is enhanced through the following innovative strategies: (1) a cross-view transfer learning framework based on dynamic anchor box optimization is designed, and the parameters of the front spark detection model YOLOv8 are transferred to the side and 45°-angle detection tasks; (2) an attention-guided feature alignment module is introduced to alleviate the feature distribution shift caused by view differences; and (3) a curriculum learning strategy is adopted, where the datasets of different views are trained separately first and then sampled to reconstruct the dataset for further training, gradually increasing the weight of samples from complex views. The experimental results show that on the self-built multi-view dataset (containing 3000 annotated images), this method achieves an average detection accuracy of 98.7%, which is 14.2% higher than that of the original YOLOv8 model. The inference speed reaches 55 FPS on an NVIDIA RTX 4090, meeting the requirements of industrial online monitoring. The research results provide key technical support for the intelligent prediction of the material removal rate in the precision machining of blades and have the potential for rapid deployment in industrial scenarios.

## 1. Introduction

In the field of high-end manufacturing, belt grinding—as a core process for the machining of precision parts—plays a crucial and vital role in industries such as aerospace and automotive manufacturing [1]. The morphology of the sparks generated during the grinding process is closely related to the wear state of the grinding wheel, the characteristics of the workpiece material, and the processing parameters [2]. Therefore, detecting the characteristics of sparks in real time and with high precision is of great significance for optimizing processing parameters, preventing equipment failures, and ensuring product quality [3]. However, the traditional spark detection method relying on manual experience places excessive dependence on the subjective judgment of operators, and it has become increasingly difficult to meet the stringent requirements of intelligent manufacturing for automation and high precision [4].

Early related research mainly employed image processing techniques based on threshold segmentation, achieving detection by extracting low-level features of sparks, such as color and brightness [5]. For instance, Li et al. [6] used HSV color space segmentation combined with morphological filtering to preliminarily facilitate the identification of grinding sparks. However, this method is extremely sensitive to changes in illumination and background noise, and the false detection rate is as high as 23% under complex working conditions. With the rise of machine learning techniques, classification methods based on support vector machines (SVMs) have been introduced into the field of spark detection [7]. By means of manual feature extraction methods such as Histogram of Oriented Gradients (HOG) and Local Binary Patterns (LBPs), certain achievements have been made in specific scenarios. Nevertheless, due to their limited feature representation ability, the generalization of such methods is not satisfactory.

In the field of computer vision, deep learning methods have been widely applied in image detection. In the paper “Advances in Image Detection Algorithms and Their Applications” [8], Zhao et al. comprehensively reviewed the development process of image detection algorithms. Starting from feature detection methods, such as the use of Haar features in combination with an Adaboost classifier for object detection, to a series of new algorithms based on convolutional neural networks (CNNs) in deep learning, they clearly depicted the development context of this field. Object detection algorithms based on CNNs, such as the R-CNN series, the YOLO series, and the Single-Shot MultiBox Detector (SSD), have significantly improved the accuracy and efficiency of object detection due to their powerful automatic feature extraction capabilities. These algorithms have already become mainstream technologies in image detection.

The YOLO series of algorithms, with their unique one-stage detection architecture, have successfully achieved a good balance between real-time performance and detection accuracy and are highly favored by researchers and engineering application personnel [1,2,3,4,5]. In the paper “Improved YOLOv5 Algorithm for Object Detection in Complex Environments” [1], Smith et al. proposed an improvement strategy for YOLOv5. Through optimizing the network structure (for instance, by adjusting the parameters and connection methods of convolutional layers), improving the feature extraction module, and introducing an attention mechanism to enhance the capture of key features, the object detection performance of the algorithm in complex scenarios was significantly improved. In the paper “YOLOv8-Based Real-Time Object Detection for Autonomous Vehicles”, Garcia et al. [2] applied YOLOv8 to a real-time object detection task considering autonomous vehicles. Aiming at the particularity of an autonomous driving scenario, the YOLOv8 model was optimized, including adjusting the size of anchor boxes to fit the shapes of different objects such as vehicles, pedestrians, and traffic signs, providing strong support for the environmental perception of the autonomous driving system.

Transfer learning, as a key technology in the field of deep learning, involves utilizing models that are pre-trained on large-scale general datasets (such as ImageNet and COCO) to quickly adapt to new tasks and specific domain datasets, effectively solving the problems of insufficient data volume and excessive training time in new tasks [9,10]. In the paper “Transfer Learning in Computer Vision: A Comprehensive Survey”, Zhang et al. [9] deeply analyzed the principles, methods, and rich application scenarios of transfer learning in the field of computer vision. Various transfer learning strategies, from model fine-tuning and feature transfer to domain adaptation, provide flexible and efficient solutions for image analysis tasks in different scenarios. In the field of intelligent manufacturing, in the paper “Application of Transfer Learning in Intelligent Manufacturing for Image Recognition”, Li et al. [10] applied transfer learning and used models pre-trained on large-scale image datasets such as ImageNet for image recognition tasks regarding industrial products. By fine-tuning some layers of the model and training on a small number of industrial product image samples, the accuracy of the model for tasks such as product defect and model identification was improved, and the training time was shortened compared with training from scratch, fully demonstrating the great advantages of transfer learning in improving the efficiency of model training and generalization ability.

Xia Y and his co-authors proposed a method that combines the Pyramid Convolution U-Net (PC-Unet) with Active Learning (AL) for change detection between optical and Synthetic Aperture Radar (SAR) images. Specifically, PC-Unet integrates pyramid convolution within a four-stage U-Net framework to capture multi-scale information, thereby enhancing detection performance. Additionally, it reduces the cost of obtaining labeled data. The experimental results demonstrate that this method outperforms a variety of state-of-the-art unsupervised and supervised heterogeneous remote sensing change detection methods [11]. Yu H. and his colleagues proposed a method based on the enhanced YOLOV8 model. By integrating the Convolutional Block Attention Module (CBAM) attention mechanism into the backbone network, the extraction of small object features was strengthened. The Path Aggregation Network (PANet) was replaced with the Weighted Bidirectional Feature Pyramid Network (BiFPN) to optimize the feature fusion process. This improvement gave priority to the small object features within the deep features and also facilitated the fusion of multi-scale features. The results show that, compared with the YOLOV8 model, the improved model achieved significant performance enhancements in terms of metrics such as precision, recall rate, and mean average precision (mAP) [12].

In research on solving the domain shift problem, Yuhua Chen and his colleagues achieved significant results in the paper “Domain Adaptive Faster R-CNN for Object Detection in the Wild” [13]. Based on the Faster R-CNN model, they conducted domain adaptation at both the image level and the instance level, designed corresponding domain adaptation components, and learned a domain-invariant Region Proposal Network (RPN) through consistency regularization. This method was experimentally tested on multiple datasets, effectively demonstrating its effectiveness in different domain shift scenarios and providing important references for cross-domain object detection. Nevertheless, there are still some challenges in existing methods. Wuyang Li and his co-authors pointed out in the paper “SIGMA: Semantic-complete Graph Matching for Domain Adaptive Object Detection” [14] that current category-level adaptation methods overlook the significant variance within categories and inadequately handle the semantics of domain mismatches in training batches. They proposed the SIGMA framework. By generating hallucinated nodes through the Graph Embedding Semantic Completion module (GSC) to complete mismatched semantics, they reframed the domain adaptation problem as a graph matching problem and achieved fine-grained domain alignment using the Bipartite Graph Matching Adapter (BGM). The experimental results show that SIGMA significantly outperforms existing methods in multiple benchmarks. These two papers are of great significance in the research of cross-domain object detection and lay a solid foundation for subsequent research work. Based on the existing research, this paper will further explore more effective cross-domain object detection methods to address the complex scenarios in practical applications.

In research on the abrasive belt grinding process, the precise monitoring of the grinding process is key to ensuring processing quality, improving production efficiency, and guaranteeing the safe operation of equipment. In the paper “A Vision-Based Monitoring System for Abrasive Belt Grinding Process” [6], Yang et al. constructed a vision-based monitoring system for the abrasive belt grinding process. This system collects images through a camera installed near the grinding area, uses image preprocessing techniques to remove noise and interference, and then extracts key features in the images, such as the brightness and texture of the grinding area, realizing the real-time monitoring of information such as the state of the abrasive belt and the surface quality of the workpiece during the grinding process, laying the hardware and algorithm foundation for the subsequent optimization control of the grinding process based on image analysis. In the paper “Analysis of Spark Image Characteristics in Abrasive Belt Grinding and Its Application” [7], Wang et al. focused on the characteristics of abrasive belt grinding spark images. Through a large number of experiments, they collected spark images under different grinding parameters, used image processing and analysis techniques to extract characteristic parameters of sparks (e.g., length, brightness, and frequency), and established a model of the relationship between these characteristics and grinding parameters (e.g., grinding speed and feed rate), as well as the wear state of the abrasive belt, providing an important theoretical basis for the monitoring and fault diagnosis of the grinding process based on spark images. In addition, in the paper “Monitoring Abrasive Belt Wear Based on Spark Image Analysis” [8], Chen et al. proposed a method for monitoring the wear of abrasive belts based on spark image analysis. By analyzing the changing characteristics of spark images, such as the increase in the irregularity of the spark morphology and the change in the brightness distribution, they constructed a wear assessment model which can predict the wear degree of the abrasive belt relatively accurately, providing an effective technical means for the timely replacement of the abrasive belt and the avoidance of any decline in processing quality. In the paper “An Intelligent Detection Method for Abrasive Belt Grinding Sparks Based on Deep Learning” [9], Zhou et al. used deep learning technology to construct a model for the detection of abrasive belt grinding sparks based on a convolutional neural network. Compared with the traditional detection methods based on manual features, this model can automatically learn the complex features in spark images, and the detection accuracy was improved. In the paper “Feature Extraction of Abrasive Belt Grinding Spark Images for Process Monitoring” [10], Deng et al. focused on the examination of a feature extraction method for abrasive belt grinding spark images and proposed a variety of feature extraction algorithms for spark images, such as those based on Local Binary Patterns (LBPs) and the Histogram of Oriented Gradients (HOG), effectively improving the ability to extract key features in spark images and providing more reliable data support for subsequent process monitoring and analysis.

Although certain research achievements have been made in the detection of abrasive belt grinding sparks from images, there is still significant room for improvement in key performance indicators such as the detection accuracy, real-time performance, and model generalization ability of the existing methods. On one hand, factors characterizing complex grinding environments, such as strong light interference and dust pollution, will seriously affect the quality of spark images, resulting in a decrease in the accuracy of detection methods. On the other hand, with the continuous increase in the requirements of industrial production for the automation and intelligence of the grinding process, the detection system is required to have higher real-time performance and stronger generalization ability to adapt to detection tasks under different working conditions. This study innovatively introduces YOLOv8 transfer learning technology into the field of abrasive belt grinding spark image detection, giving full play to the high efficiency of YOLOv8 in object detection tasks and the good adaptability of transfer learning to small-sample and domain-specific tasks. It is committed to greatly improving the accuracy and real-time performance of abrasive belt grinding spark detection, providing advanced technical support for the intelligent monitoring and precise control of the abrasive belt grinding process. At the same time, referring to the idea of dynamic anchor boxes in DAB-DETR proposed by Liu et al. [15], this study actively explores the optimization path of the YOLOv8 detection mechanism, hoping to further tap into the potential of the model and significantly improve detection performance.

The overall structure of the paper is shown in Figure 1. First, Section 1 presents the research background and objectives. Subsequently, Section 2 unfolds from four dimensions: constructing a multi-view spark image dataset to provide a data foundation for subsequent research; designing a cross-view transfer learning framework based on dynamic anchor box optimization to enhance model performance; utilizing an attention-guided feature alignment module to strengthen feature processing capabilities; and adopting a curriculum learning strategy to optimize the learning process.

Next is Section 3, which includes building the experimental platform and environment to provide physical support for the experiments; setting up comparative experiments to highlight the advantages of the research methods; and determining evaluation indicators to measure experimental effects [16]. In Section 4, operations such as training side-view spark images with YOLOv8, transferring the side-view spark training model to front-view spark images, and iterative training with the curriculum learning strategy are carried out in sequence, and the experimental results are ultimately presented. Finally, in Section 5, the research achievements are summarized, the significance and value of the research are expounded, and the directions for subsequent research are prospected [17].

This study proposes a multi-perspective spark image detection method based on YOLOv8 transfer learning.

During the research process, a multi-pose spark image dataset covering front, side, and 45° angle views was constructed, laying a data foundation for subsequent research. A cross-view transfer learning framework optimized by dynamic anchor boxes was designed [18]. The parameters of the front-view spark detection model YOLOv8 were transferred to detection tasks of other views, improving the detection performance under different views. An attention-guided feature alignment module was introduced to effectively mitigate the feature distribution shift caused by view differences, enhancing the model’s adaptability to image features from different perspectives. A curriculum learning strategy was adopted. First, datasets of different views were trained separately, and then the dataset was reconstructed through sampling for further training. The weight of samples from complex views was gradually increased, allowing the model to better learn complex image features [19].

To verify the effectiveness of the method, comparison experiments with multiple models were carried out on the self-built dataset. The results show that this method performs outstandingly, providing key technical support for the intelligent prediction of the material removal rate in precision blade machining. It has the potential for rapid deployment in industrial scenarios and is expected to promote the intelligent development of the blade processing industry.

## 2. Methods

### 2.1. Construction of the Multi-View Spark Image Dataset

To achieve cross-view spark image detection, in this study, a spark image dataset with three postures was constructed. Spark images were synchronously collected from three different views on the abrasive belt grinding experimental platform: the front view (front spark image), the left view (side spark image), and the horizontal 45°angle direction. The experiments covered a variety of combinations of grinding parameters, including different grinding speeds, feed rates, and types of workpiece materials (the experimental planning parameters are listed in Table 1). A total of 3000 original images were collected. Through a strict data screening and annotation process, samples with blurriness, excessive noise, and inaccurate annotations were removed. Finally, a high-quality dataset containing 3000 annotated images was obtained. As shown in Figure 2, the annotation information precisely defines key features such as the position, shape, and category of the sparks, providing a solid data foundation for subsequent model training and evaluation.

When grinding the workpiece with a three-axis machine tool, we selected GCr15 with a hardness of HRC58 and dimensions of 170 mm × 41 mm × 50 mm. The average roughness of GCr15 was 0.2 μm. The abrasive belt was installed on the *Z*-axis through the driving wheel, the tensioning wheel, and the contact wheel. When the abrasive belt grinded the workpiece, the abrasive belt speed was set within the range of 26–38 m/s, with an increment of 2 m/s for each stroke. The workpiece feed speed was within the range of 1.5–4.5 mm/s, with an increment of 0.5 mm/s for each stroke. The theoretical grinding depth was 0.3 μm, and the actual grinding depth was measured and recorded with a profilometer after the end of each stroke. The belt material was corundum, with a width of 20 mm. When rotating at a high speed, it cut the workpiece well and generated a spark field. A laptop was used to synchronously collect and record the spark images in real time.

Figure 2a is a spark image at a horizontal 45-degree angle annotated with Labelme. Figure 2b is a front-view spark image annotated with Labelme, and Figure 2c is a left-view spark image annotated with Labelme. In the figures, the spark regions are marked with rectangular boxes, labeled with the tag “fire”, and corresponding JSON files were generated, which were used for the detection of the target regions of the spark images in the later stage.

### 2.2. The Design of the Cross-View Transfer Learning Framework Based on Dynamic Anchor Box Optimization

#### 2.2.1. Generation of Dynamic Anchor Boxes

Drawing on the idea of dynamic anchor boxes in DAB-DETR proposed by Liu et al. [15], this study optimizes the anchor box generation mechanism of YOLOv8. In the traditional YOLOv8, the size and ratio of anchor boxes are fixed, making it difficult to adapt to the diversity of the shapes and sizes of sparks under different views. This method dynamically generates anchor boxes according to the statistical characteristics of sparks under different views [20]. Specifically, through statistical analysis of the width, height, and aspect ratio of sparks from different views in the training dataset, the optimal size and ratio range of anchor boxes for each view are determined. During the model training process, according to the view information of the current input image, the parameters of the anchor boxes are dynamically adjusted, enabling them to more closely fit the actual shapes of sparks under different views, thus improving the detection accuracy of the model for sparks from different views.

(1)The determination of the optimal size and ratio range through statistical analysis

For each view in the training dataset, the width (*w*) and height (*h*) data of all sparks from that view were collected. Statistical measures of the width and height, including the mean ($\bar{w}$, $\bar{h}$) and the standard deviation ($\sigma_{w}$, $\sigma_{h}$), were collected.

The aspect ratio (*r* = *w*/*h*), as well as its mean ($\bar{r}$) and standard deviation ($\sigma_{r}$), was calculated. The method for calculating the mean value of the width is shown in Formula (1):(1)$$\bar{w}=\frac{1}{n}\sum_{i=1}^{n}w_{i}$$

In Formula (1), $\bar{w}$ is the mean value of the width, *w_i_* is the width of the *i*-th anchor box, and a total of *n* samples are counted.

The method for calculating the mean value of the height is shown in Formula (2):(2)$$\bar{h}=\frac{1}{n}\sum_{i=1}^{n}h_{i}$$

In Formula (2), $\bar{h}$ is the mean value of the height, and *h_i_* is the height of the *i*-th anchor box.

The method for calculating the mean value of the aspect ratio is shown in Formula (3):(3)$$\bar{r}=\frac{1}{n}\sum_{i=1}^{n}\frac{w_{i}}{h_{i}}$$

In Formula (3), $\bar{r}$ is the mean value of the aspect ratio, and the definitions of *w_i_* and *h_i_* are the same as those in Formulas (1) and (2). The method for calculating the standard deviation of the width is shown in Formula (4):(4)$$\sigma_{w}=\sqrt{\frac{1}{n}\sum_{i=1}^{n}{\left(w_{i}-\bar{w}\right)}^{2}}$$

In Formula (4), $\sigma_{w}$ is the standard deviation of the width, *w_i_* is the width of the *i*-th anchor box, $\bar{w}$ is the mean value of the width, and a total of *n* samples are calculated.

The method for calculating the standard deviation of the height is shown in Formula (5):(5)$$\sigma_{h}=\sqrt{\frac{1}{n}\sum_{i=1}^{n}{\left(h_{i}-\bar{h}\right)}^{2}}$$

In Formula (5), $\sigma_{h}$ is the standard deviation of the height, *h_i_* is the height of the *i*-th anchor box, and $\bar{h}$ is the mean value of the height.

The method for calculating the standard deviation of the aspect ratio is shown in Formula (6):(6)$$\sigma_{r}=\sqrt{\frac{1}{n}\sum_{i=1}^{n}{\left(\frac{w_{i}}{h_{i}}-\bar{r}\right)}^{2}}$$
where $\sigma_{r}$ is the standard deviation of the aspect ratio, *w_i_* is the width of the *i*-th anchor box, *h_i_* is the height of the *i*-th anchor box, and $\bar{r}$ is the mean value of the aspect ratio.

The width range of the anchor box can be set as [$\bar{w}$ − *k*$\sigma_{w}$, $\bar{w}$ + *k*$\sigma_{w}$]. The height range is set as [$\bar{h}$ − *k*$\sigma_{h}$, $\bar{h}$ + *k*$\sigma_{h}$].

The aspect ratio range is set as [$\bar{r}$ − *k*$\sigma_{r}$, $\bar{r}$ + *k*$\sigma_{r}$]. k is an adjustable coefficient, and the value is usually 1 or 2.

(2)Dynamic adjustment of the parameters of the anchor boxes

During the model training process, according to the view information of the input image, the appropriate values from the size and ratio range of the anchor boxes corresponding to that view were selected. In Formula (6), $\sigma_{r}$ is the standard deviation of the aspect ratio, *w_i_* is the width of the *i*-th anchor box, *h_i_* is the height of the *i*-th anchor box, and $\bar{r}$ is the mean value of the aspect ratio.

The width range of the anchor box was set as [$\bar{w}$ − *k*$\sigma_{w}$, $\bar{w}$ + *k*$\sigma_{w}$]. The height range was set as [$\bar{h}$ − *k*$\sigma_{h}$, $\bar{h}$ + *k*$\sigma_{h}$]. The aspect ratio range was set as [$\bar{r}$ − *k*$\sigma_{r}$, $\bar{r}$ + *k*$\sigma_{r}$]. Here, *k* is an adjustable coefficient, and its value is usually 1 or 2. Appropriate anchor box parameters were randomly selected from the range. Suppose the width range under the current view is [*w_min_*,*w_max_*], the height range is [*h_min_*,*h_max_*], and the aspect ratio range is [*r_min_*,*r_max_*]. The width was randomly selected as *w_anchor_* = random(*w_min_*,*w_max_*), the aspect ratio was randomly selected as *r_anchor_* = random(*r_min_*,*r_max_*), and the height was calculated as *h_anchor_* = (*w_anchor_*)/*r_anchor_*.

#### 2.2.2. Cross-View Transfer Learning Strategy

In this study, a cross-view transfer learning framework based on dynamic anchor box optimization was designed. Firstly, the YOLOv8 model was pre-trained on the front spark image dataset to obtain a front spark detection model with good performance. Then, the parameters of this model were transferred to the side and 45° angle detection tasks. During the transfer process, the parameters of the backbone network of the pre-trained model were kept fixed, and only the parameters of the detection head part were fine-tuned. In this way, the advantages of the front spark detection model in feature extraction were fully utilized to accelerate the convergence speed of the side and 45° angle detection models, and at the same time, the number of samples required for training was reduced. In addition, combined with the dynamic anchor box optimization mechanism, this model can better adapt to the feature differences of sparks under different views, further improving the cross-view detection performance. The corresponding specific transfer strategy is shown in Figure 3.

Figure 3 describes the process of using five different models of YOLOv8, namely YOLOv8n, YOLOv8s, YOLOv8m, YOLOv8L, and YOLOv8x, for image training and model optimization. Firstly, these models were used to train the front-facing spark images. Then, the training results of the front-facing spark images were transferred to the training of the side-facing spark images. After that, the training results of the side-facing spark images were transferred to the training of the 45-degree spark images. Finally, by adopting the curriculum learning strategy and going through multiple iterative trainings, the optimal model was obtained.

The advantages of transferring the image training results based on YOLOv8 are as follows:

Improve training efficiency: There is no need to train the model completely from scratch every time for spark images at new angles. By making use of the existing training approaches, the training time can be significantly shortened, and the consumption of computing resources can be reduced. For example, the model trained on front-facing spark images first has already learned some basic features of sparks. When it is subsequently transferred to angles such as the side, the model can quickly adapt to the new data on this basis, avoiding repeated learning.

Reduce data requirements: For some angles where it is difficult to obtain a large amount of labeled data (such as the side and 45-degree views), the training results of relatively abundant data from other angles (such as the front view) can be utilized to reduce the dependence on the amount of data at specific angles. For instance, in practical scenarios, it may be difficult to obtain a large number of high-quality side-facing spark images, and transfer learning can alleviate the problem of insufficient data.Enhance the model’s generalization ability: Through the transfer and iterative optimization of the training results of images at different angles, the model can learn more comprehensive features of sparks. As a result, when faced with spark detection tasks in various practical scenarios, the model has better performance and adaptability.Achieve knowledge sharing: The knowledge learned during the training processes at different angles can be transmitted to each other, enabling the model to understand spark images from multiple perspectives and enhancing its ability to recognize spark targets.

Although they are all spark images, there are perspective differences among the front-facing, side-facing, and 45-degree images. The knowledge learned by the model at the source angle (such as the front view) may not be fully applicable to the target angle (such as the side view), resulting in poor transfer effects. For example, the light and shadow as well as contour features of side-facing sparks are quite different from those of front-facing sparks, which may cause deviations in the model’s recognition. Therefore, we introduced attention-guided features to the relevant modules to eliminate these impacts.

### 2.3. Feature Alignment Module Guided by Attention

#### 2.3.1. Module Design

To alleviate the problem of feature distribution shift caused by perspective differences, this study introduces a feature alignment module guided by attention. This module is located after the feature fusion layer of the YOLOv8 model. By learning the correlations between feature maps from different perspectives, it automatically adjusts the weights of features, enabling the model to pay more attention to the features that make significant contributions to cross-perspective detection. Specifically, the module first performs a global average pooling operation on the feature maps from different perspectives, compressing the feature maps into one-dimensional vectors to obtain the global feature representation of each feature map. Then, by calculating the similarity between these global feature vectors, an attention matrix is constructed. This matrix reflects the similarity degree between the feature maps from different perspectives. The higher the similarity, the greater the weight of the corresponding feature in the subsequent feature fusion process. Finally, the attention matrix is weighted and fused with the original feature maps to obtain the feature maps after feature alignment processing, which are then fed into the subsequent detection head for object detection.

Suppose there is a set of feature maps {*F*_1_, *F*_2_, …, *F_n_*} from different perspectives, where *F_i_* represents the feature map of the *i*-th perspective, and its dimension is *C* × *H* × *W* (*C* is the number of channels, *H* is the height, and *W* is the width). The feature maps are obtained through the following three steps.

1. Global Average Pooling: A global average pooling operation is performed on each feature map *F_i_* to compress it into a one-dimensional vector, as shown in Formula (7).(7)$$g_{i}=\frac{1}{H\times W}\sum_{h=1}^{H}\sum_{w=1}^{W}F_{i}(:,h,w)$$

In Formula (7), *g_i_* represents the global feature representation of the feature map from the *i*-th perspective, and its dimension is *C* × 1.

2. Calculate the similarity and construct the attention matrix: The similarity between any two global feature vectors is calculated. The formula is as follows:(8)$$A_{ij}=\frac{g_{i}^{T}g_{j}}{\left||g_{i}||||g_{j}|\right|}$$

In Formula (8), *A_ij_* is an element of the attention matrix *A*, representing the similarity between the feature map of the *i*-th perspective and that of the *j*-th perspective. ||*g_i_*|| and ||*g_j_*|| are the norms of the vectors *g_i_* and *g_j_*, respectively. The dimension of the attention matrix *A* is *n* × *n*.

3. Weighted Fusion: For each feature map *F_i_*, a weighted summation of all feature maps is performed according to the elements in the *i*-th row of the attention matrix. The calculation method is shown in Formula (9).
(9)$$F_{i}^{\prime }=\sum_{j=1}^{n}A_{ij}F_{j}$$ where $F_{i}^{\prime }$ is the feature map of the *i*-th perspective after feature alignment processing. The finally obtained feature map set {$F_{1}^{\prime }$, $F_{2}^{\prime }$, …, $F_{n}^{\prime }$} can be fed into the subsequent detection head for object detection.

#### 2.3.2. The Role of the Feature Alignment Module Guided by Attention

The core role of the feature alignment module guided by attention is to enhance the model’s ability to extract common features under different perspectives, while suppressing the feature deviations caused by perspective differences. Through introducing the attention mechanism, the model can automatically identify and highlight the features that play a crucial role in cross-perspective detection, thereby improving the consistency and discriminability of the features. For example, under different perspectives, features such as the brightness and color of sparks may change, but the basic shape and contour features of sparks have a certain degree of stability. The feature alignment module guided by attention can, through learning the correlations between feature maps from different perspectives, strengthen the extraction of these stable features and simultaneously reduce the influence of interfering features caused by perspective changes. In this way, it can effectively im-prove the performance of the model in cross-perspective detection tasks.

### 2.4. Curriculum Learning Strategy

#### 2.4.1. Strategy Implementation Steps

This study adopted the curriculum learning strategy to optimize the model training process. The specific steps are as follows: first, the datasets were trained from different perspectives separately, enabling the model to initially learn the feature patterns of sparks under each perspective. At this stage, the model focused on feature extraction and classification of single-perspective data, and converged quickly and achieve a certain level of detection accuracy. Then, the datasets of each perspective were sampled, and a mixed dataset was reconstructed according to a certain proportion. In the mixed dataset, the weight of samples from complex perspectives was gradually increased. For example, some representative and feature-complex samples from the side and 45°angle perspective samples were selected, and their frequency of appearance in the mixed dataset was increased. In this way, the model was guided to gradually adapt to the feature differences between different perspectives, enhancing the model’s generalization ability. During the training process, the weights of samples from different perspectives in the mixed dataset were dynamically adjusted according to the model’s performance, such that the model could maintain a steady improvement in detection accuracy while continuously learning complex samples.

#### 2.4.2. Strategy Advantages

The advantage of the curriculum learning strategy is that allowed the model to obtain knowledge at different difficulty levels step by step, avoiding learning difficulties and overfitting problems caused by facing overly complex data in the early stage of training. Through single-perspective training first, the model quickly mastered the basic features of sparks under each perspective and established the initial ability of feature extraction and classification. Subsequently, by introducing a mixed dataset and increasing the weight of complex-perspective samples, the model was gradually exposed to and adapted to the differences between different perspectives, effectively improving the model’s generalization ability. This strategy not only improved the model’s training efficiency but also enabled the model to better learn the key features required for cross-perspective detection under the condition of limited training samples, thus achieving better performance in the multi-perspective spark image detection task.

## 3. Experimental Design

### 3.1. Experimental Platform and Environment

The experiment was carried out on a computer equipped with an NVIDIA RTX 4090 GPU, and the operating system was Windows 11. The deep learning framework selected was PyTorch 1.9.0, the CUDA version was 11.1, and the CUDNN version was 8.0.5. The experimental platform was equipped with a belt grinding experimental device that could precisely control processing parameters such as the grinding speed and feed rate and it was equipped with a high-definition camera to collect spark images from different perspectives.

### 3.2. Setup of Comparative Experiments

To comprehensively evaluate the performance of the multi-perspective spark image detection method based on YOLOv8 transfer learning proposed in this paper, the following comparative experiments were set up:

The original YOLOv8 model was directly trained on the multi-perspective spark image dataset without using any transfer learning or optimization strategies, serving as a benchmark model for comparison with the proposed method.

Without using the curriculum learning strategy, the YOLOv8 transfer learning model was directly trained on the mixed dataset to compare the improvement effect of the curriculum learning strategy on the model’s training efficiency and generalization ability.

### 3.3. Evaluation Indicators

The mean average precision (mAP), recall rate, precision rate, and loss rate were used as evaluation indicators. The mean average precision comprehensively reflects the model’s detection accuracy for various types of targets under different thresholds. The recall rate measures the model’s ability to detect all real targets. The precision rate represents the proportion of truly correct targets among the targets detected by the model. The inference speed reflects the real-time performance of the model in practical applications. These indicators can comprehensively and objectively evaluate the performance of the model in the multi-perspective spark image detection task.

In mAP50, “mAP” is the abbreviation of “mean Average Precision”, which means the mean value of average precision. The number “50” indicates that the threshold of the IoU (Intersection over Union) is 0.5. mAP50 is used to evaluate the comprehensive performance of the object detection model across different categories. It measures the matching degree between the bounding boxes predicted by the model and the ground truth bounding boxes. When the IoU is greater than or equal to 0.5, it is considered a correct prediction. mAP50 is obtained by calculating the average precision of each category and then taking the mean value. The closer its value is to 1, the better the detection performance of the model.

Since the research object of this paper has only one target, “fire”, the following calculation method was adopted. Firstly, the prediction results of the model were sorted in descending order of confidence. Then, the prediction results were traversed one by one, and the IoU of each prediction result was calculated with the ground truth bounding box. When the IoU was greater than or equal to 0.5, this prediction was regarded as a correct detection. Next, the precision was calculated at different recall rates according to the definitions of recall and precision. Recall refers to the ratio of the number of correctly detected targets to the number of real targets in this category, and precision refers to the ratio of the number of correctly detected targets to the number of predicted targets. Through calculation, a series of recall–precision pairs were obtained, and these points were connected to form a curve; AP is the area under this curve.

## 4. Experimental Results and Analysis

### 4.1. Side Spark Image Training with YOLOv8

First, we used the five different original YOLOv8 models—namely YOLOv8n, YOLOv8s, YOLOv8m, YOLOv8l, and YOLOv8x—to train side spark images. After the training was completed, the corresponding evaluation index curves were obtained. Figure 4 shows the mean average precision curve.

As can be seen from Figure 4a, as the training progresses, the mAP50 of each model gradually tends to be stable. YOLOv8n and YOLOv8m perform better in the later stage, while YOLOv8x has larger fluctuations and its final performance is slightly inferior. In Figure 4b, it can be seen that YOLOv8n is overall more prominent during the training process, demonstrating a relatively good mean average precision, and YOLOv8x still undergoes significant fluctuations.

Figure 5 presents the precision rate curve, and Figure 6 shows the recall rate curve.

As can be seen from Figure 5, in the later stage of training, the precision rate of YOLOv8n stabilizes at approximately 0.98, and that of YOLOv8l is around 0.96, which is higher than that of YOLOv8x (about 0.9). This reflects the advantages of YOLOv8n and YOLOv8l in accurately identifying targets.

As can be seen from Figure 6, in the later stage of training, the recall rate of YOLOv8s stabilizes at approximately 0.78, that of YOLOv8n is around 0.76, and that of YOLOv8m is about 0.75, indicating that these models have relatively strong capabilities in recalling targets.

Figure 7 shows the training loss rate curve.

As can be seen from Figure 7, the training loss has the following characteristics:Box_loss: At the 100th epoch, the box_loss of YOLOv8n is approximately 0.5, and that of YOLOv8s is about 0.7, which is much lower than that of YOLOv8x (about 1.3). This indicates that the small models converge faster and have lower loss in bounding box prediction during training.Cls_loss: The cls_loss of YOLOv8n drops rapidly in the early stage of training and approaches 0 at around the 10th epoch. Although YOLOv8x has large fluctuations in the early stage, it can also converge to a level close to 0 in the later stage, demonstrating the convergence of each model in classification loss.Dfl_loss: In the later stage of training, the dfl_loss of YOLOv8n is approximately 1.0, and that of YOLOv8s is about 1.2, which is lower than that of YOLOv8x (about 1.8), reflecting the advantages of small models in distribution focal loss.

As can be seen from Figure 8, the validation loss has the following characteristics:Box_loss: At the 100th epoch, the val/box_loss of YOLOv8n is approximately 1.3, which is lower than that of YOLOv8m (about 1.8) and YOLOv8x (about 2.0). This indicates that YOLOv8n has more stable bounding box prediction and smaller loss on the validation set.Cls_loss: After convergence, the val/cls_loss of YOLOv8x approaches 0, and other models also converge rapidly to an extremely low level after initial fluctuations, indicating that the classification performance of each model on the validation set is good in the later stage.Dfl_loss: After convergence, the val/dfl_loss of YOLOv8x approaches 0, and those of YOLOv8n and YOLOv8s are also at a low level, reflecting the situation of the distribution focal loss of each model on the validation set.

**Figure 8 sensors-25-02946-f008:**
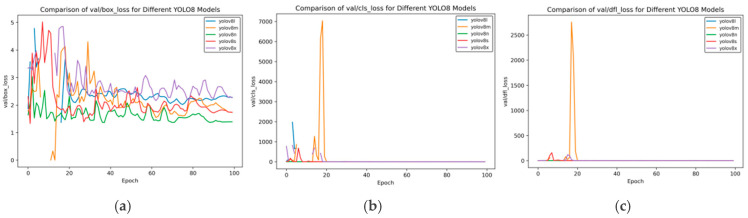
Val loss curve of YOLOv8 training for side spark images. (**a**) val box loss; (**b**) val class loss; and (**c**) val dfl loss.

Based on the comprehensive analysis of the evaluation metrics and loss functions, different YOLOv8 models have their own advantages and disadvantages in performance. Smaller models such as YOLOv8n and YOLOv8s perform well in terms of training convergence speed and some evaluation metrics, making them suitable for scenarios with limited computing resources. On the other hand, larger models such as YOLOv8x, although experiencing greater fluctuations during training, also show good potential in certain metrics and can be considered when resources are abundant and high precision is required. In practical applications, an appropriate YOLOv8 model should be selected according to the specific task requirements and available resources.

### 4.2. Transfer of the Side Spark Training Model to Frontal Spark Images

Based on the comparative analysis results of the training of side spark images using the above five models, we selected the relatively better models, namely YOLOv8n, YOLOv8s, and YOLOv8m, and transfered them to the object detection of frontal spark images. After training, the corresponding evaluation metric parameters were obtained. By comparing these with the training results of side spark images, the following set of curves was obtained.

In Figure 9, the models starting with “t1” represent the result curves of side spark object detection, and the models starting with “t2” represent the training results obtained after transferring the side spark training results to frontal spark images. Figure 9 shows the changes in performance indicators of different YOLOv8 models during the training process.

In Figure 9a, the horizontal axis is “Epoch”, and the vertical axis “metrics/mAP50 (B)” is the mean average precision when the intersection-over-union (IoU) threshold is 0.5. It can be seen from the figure that for t2 YOLOv8n and t2 YOLOv8s, they have relatively high indicator values from the beginning, and tend to be stable and approach a value of 1 in the later stage of training. This indicates that these two models can detect targets well under a relatively low IoU threshold.

For the t1 series models (t1 YOLOv8m, t1 YOLOv8n, and t1 YOLOv8s), there are large fluctuations in the early stage of training. The indicator value of t1 YOLOv8m even approaches 0. Although there is an improvement in the later stage, compared with the t2 series models, their performance still lags behind.

In Figure 9b, the horizontal axis “Epoch” represents the number of training rounds, and the vertical axis “metrics/mAP50-95 (B)” represents the mean average precision (under different IoU thresholds ranging from 0.5 to 0.95), which is an important indicator for measuring the performance of object detection models. A higher value indicates better model performance. t2 YOLOv8n and t2 YOLOv8s perform prominently in most rounds, with relatively high and stable indicator values, indicating that these two models have good detection effects within a wide range of IoU thresholds.

t1 YOLOv8m has relatively poor overall performance, with large fluctuations in the indicator values, which shows that this model has poor stability during training and its detection performance is not ideal.

Overall, the t2 series of YOLOv8 models (t2 YOLOv8n, t2 YOLOv8s, and t2 YOLOv8m) generally perform better than the t1 series models under these two indicators; the YOLOv8n and YOLOv8s models are relatively more stable and have better performance under different mAP indicators.

Figure 10 shows the comparison curve of precision after transfer learning, and Figure 11 presents the comparison curve of recall after transfer learning.

Figure 10 compares the changes in the precision indicator (precision (B)) of different YOLOv8 models during the training process (Epoch). The overall analysis is as follows:Model Classification: There are a total of six models in the figure. Both the t1 and t2 series include three versions—YOLOv8m, YOLOv8n, and YOLOv8s. These models are different variants based on the YOLOv8 object detection architecture and are commonly used in object detection tasks in images and videos.Initial Training Performance: In the initial stage of training (when the Epoch value is small), the precision of each model fluctuates significantly. Especially for the t1 series, such as t1 YOLOv8m, the precision oscillates greatly between 0 and 0.2, indicating that the model is unstable in the initial training stage and has not converged during the process of learning data features. In contrast, the t2 series models have relatively smaller fluctuations in the initial stage, showing a relatively more stable training start state.Mid-training Trend: As the Epoch increases, the precision of each model generally shows an upward trend, but there are significant differences in the upward process. Some models in the t1 series have large fluctuations during the upward process, and the precision growth is discontinuous. The t2 series models have a smoother upward process, indicating that the t2 series is more stable in the process of fitting and optimizing the data during training.Late-Stage Convergence: In the later stage of training (when the Epoch is close to 100), the precision of most models tends to be stable and approaches 1.0. This indicates that after a sufficient number of training rounds, each model can learn the features in the data well and achieve high detection precision. At the same time, the t2 series models seem to enter the stable state faster and have better stability, which means that their generalization ability may be relatively stronger.

Overall, although different YOLOv8 models can ultimately achieve high precision, the t2 series models in transfer learning perform better in terms of training stability and convergence speed.

Figure 11 shows the changes in the recall indicator (metrics/recall (B)) of different YOLOv8 models with the number of training rounds (Epoch). The comparison between the t2 models and t1 models in transfer learning is as follows:Initial Performance: In the initial stage of training, the t1 series models have large fluctuations. For example, for t1 YOLOv8m and t1 YOLOv8n, the recall rate drops rapidly from a relatively high value and then rises again, and t1 YOLOv8s fluctuates at a relatively low value. The t2 series is relatively stable, with a moderate initial value and small fluctuations.Mid-term Trend: Between Epochs 20 and 60, the recall rate of the t1 series oscillates frequently, with significant drops and rebounds occurring multiple times. Although the t2 series also has fluctuations, the overall trend is smoother and relatively more stable.Late-Stage Stability: When approaching Epoch 100, the recall rates of t2 series models such as t2 YOLOv8n and t2 YOLOv8s stabilize at a relatively high level. Among the t1 series, t1 YOLOv8m and t1 YOLOv8n still have some fluctuations, and although t1 YOLOv8s is stable, its value is relatively low.

Overall, the t2 series models perform better than the t1 series models in terms of the stability of the recall rate and the final level reached.

Figure 12 presents the training loss rate curve of transfer learning.

Figure 12 shows the changes in the bounding box loss (box_loss), classification loss (cls_loss), and distribution focal loss (dfl_loss) of different YOLOv8 models during the training process with the number of training rounds (Epochs). The following is a comparative analysis of the t2 and t1 models:


**Bounding Box Loss (box_loss)**


Initial Loss: In the initial stage of training, the box_loss value of t1 YOLOv8m is the highest, starting above 3.0; the initial values of the t2 series are relatively lower, with t2 YOLOv8m starting at approximately 2.4.Rate of Decrease: t1 YOLOv8m decreases rapidly in the early stage but has large overall fluctuations; the t2 series decreases relatively steadily. For example, t2 YOLOv8s has small fluctuations during training.Final Loss: When approaching Epoch 100, the final loss of t1 YOLOv8n is the lowest, and that of t1 YOLOv8m is relatively high; the t2 series as a whole is relatively close, with t2 YOLOv8s being slightly lower.


**Classification Loss (cls_loss)**


Initial Loss: The initial cls_loss of t1 YOLOv8m is extremely high, exceeding 17.5; t1 YOLOv8s is approximately 5.0, and the t2 series is below 2.5. The t2 series performs better initially.Rate of Decrease: t1 YOLOv8m decreases extremely rapidly in the early stage, and t1 YOLOv8s also decreases quickly; the t2 series decreases at a slower but stable rate.Final Loss: In the later stage, the cls_loss of each model is low and close, with the t2 series being slightly lower and having slightly better stability.


**Distribution Focal Loss (dfl_loss)**


Initial Loss: The initial dfl_loss of t1 YOLOv8m exceeds 3.0, which is the highest; the t2 series is around 2.5, and t2 YOLOv8n is the lowest.Rate of Decrease: t1 YOLOv8m decreases rapidly in the early stage, accompanied by large fluctuations; the t2 series decreases relatively steadily, and t2 YOLOv8s has the smallest fluctuations.Final Loss: Near Epoch 100, the final loss of t1 YOLOv8n is relatively low, and that of t1 YOLOv8m is relatively high; the t2 series as a whole is close, with t2 YOLOv8n being slightly lower.

Overall, some models in the t1 series, such as t1 YOLOv8m, have a rapid decrease in loss in the early stage but with large fluctuations; the t2 series has a lower initial loss, better stability during the training process, and relatively stable final performance in different types of losses.

Figure 13 shows the changes in the distribution focal loss (val/dfl_loss), bounding box loss (val/box_loss), and classification loss (val/cls_loss) of different YOLOv8 models on the validation set with the number of training rounds (Epoch). The comparison between the t2 and t1 models is as follows:


**Distribution Focal Loss (val/dfl_loss)**


Initial Fluctuations: In the initial stage of training, t1 YOLOv8m shows an extremely high loss value, exceeding 2500, and t1 YOLOv8s also has a short-term fluctuation; the t2 series is stable as a whole, and the loss values of t2 YOLOv8m, t2 YOLOv8n, and t2 YOLOv8s are always close to 0.Later-Stage Trend: As the Epoch increases, except for the initial fluctuations, the t1 series tends to approach 0 subsequently; the t2 series always remains at an extremely low level, and its stability is significantly better than that of the t1 series.

**Figure 13 sensors-25-02946-f013:**
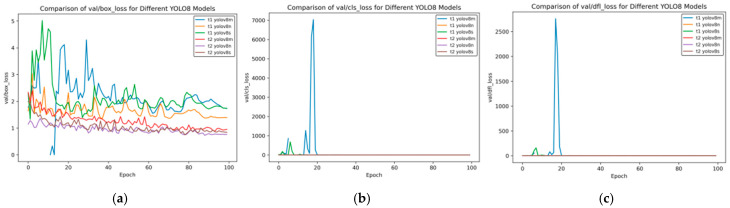
Val loss curve of transfer training. (**a**) Val box loss; (**b**) val class loss; and (**c**) val dfl loss.


**Bounding Box Loss (val/box_loss)**


Initial Performance: At the beginning of training, the t1 series models have large fluctuations. The loss values of t1 YOLOv8m and t1 YOLOv8s fluctuate greatly below 5; the t2 series is relatively stable with a lower initial value.Later-Stage Changes: When approaching Epoch 100, although the loss values of the t1 series gradually decrease, they still fluctuate around 1–2; the t2 series is generally below 1, and t2 YOLOv8s has the lowest value. The t2 series performs better as a whole.


**Classification Loss (val/cls_loss)**


Initial Fluctuations: In the initial stage, t1 YOLOv8m shows an extremely high loss, exceeding 7000, and t1 YOLOv8s also has fluctuations; the t2 series always remains stable at a level close to 0.Later-Stage Trend: As the training progresses, except for the large initial fluctuations, the t1 series gradually decreases; the t2 series always remains extremely low and stable, and the t2 series far surpasses the t1 series in terms of the stability of the classification loss.

Overall, in terms of the performance of the loss on the validation set, the t2 series models are more stable than the t1 series, and the loss values are generally lower. In particular, they have clear advantages in val/dfl_loss and val/cls_loss.

### 4.3. Iterative Training with the Curriculum Learning Strategy

After the first transfer learning training, we observed a significant improvement in the model’s performance compared to that before the transfer. Inspired by this, we further expanded the application of transfer learning by applying the model trained through the transfer of frontal spark images to the training of 45-degree spark images. Through in-depth analysis of the experimental data, the results show that the performance of the transferred model is also significantly better than that before the transfer.

It is worth noting that the datasets involved in these two transfer processes are highly independent. Specifically, the side spark object detection dataset only includes side spark images, the frontal spark object detection dataset only contains frontal spark images, and the 45-degree spark object detection dataset only stores 45-degree spark images.

To further enhance the complexity and generalization ability of the model, we introduced the curriculum learning strategy, fused the above three different types of datasets according to a specific proportion, and used the latest transfer learning model for further training. After multiple rounds of experiments and parameter adjustments, we finally successfully obtained the optimal model.

In the transfer learning training, we selected three models, namely YOLOv8n, YOLOv8s, and YOLOv8m, and carried out the training using the curriculum learning strategy. Through comparative evaluation, the model obtained after multiple transfer training based on the YOLOv8m model shows the best performance. When comparing the performance indicators of this model with those of the model initially used for side spark image detection, significant improvements have been made across all indicators. The following set of curves intuitively confirms this conclusion.

In Figure 14, Figure 15, Figure 16, Figure 17 and Figure 18, t6_yolov8m represents the model after multiple transfer learning trainings, and YOLOv8m represents the model without transfer learning. The figures display the comparison of various indicators of the two models, YOLOv8m and t6_yolov8m, during the training and validation processes. The comprehensive analysis is as follows:Loss Indicators: For various loss indicators during training and validation (train/box_loss, train/cls_loss, train/dfl_loss, val/box_loss, val/cls_loss, and val/dfl_loss), both the initial and final values of t6_yolov8m are generally lower than those of YOLOv8m. Especially in the early stage of training, extremely high peaks appear in train/cls_loss, train/dfl_loss, val/cls_loss, and val/dfl_loss of YOLOv8m, while t6_yolov8m remains relatively stable. This indicates that t6_yolov8m converges faster and the training process is more stable.Performance Indicators: In terms of performance indicators such as metrics/mAP50 (B), metrics/mAP50-95 (B), metrics/precision (B), and metrics/recall (B), t6_yolov8m significantly outperforms YOLOv8m. The indicator values of t6_yolov8m are higher and have smaller fluctuations, while YOLOv8m has large fluctuations in the early stage and lower overall values. This shows that t6_yolov8m has better detection accuracy and stability.

**Figure 14 sensors-25-02946-f014:**
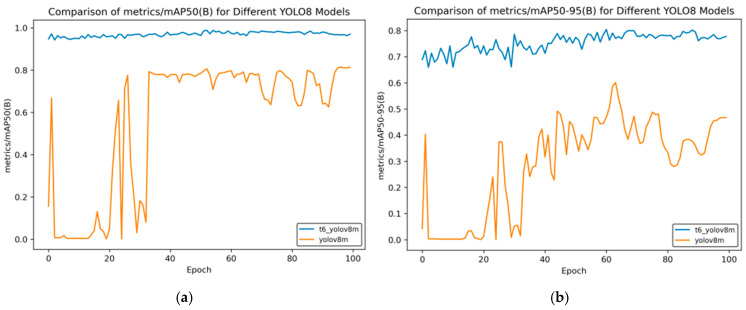
mAP comparison curve after course learning strategy. (**a**) mAP50 curve; (**b**) mAP50-95 curve.

**Figure 15 sensors-25-02946-f015:**
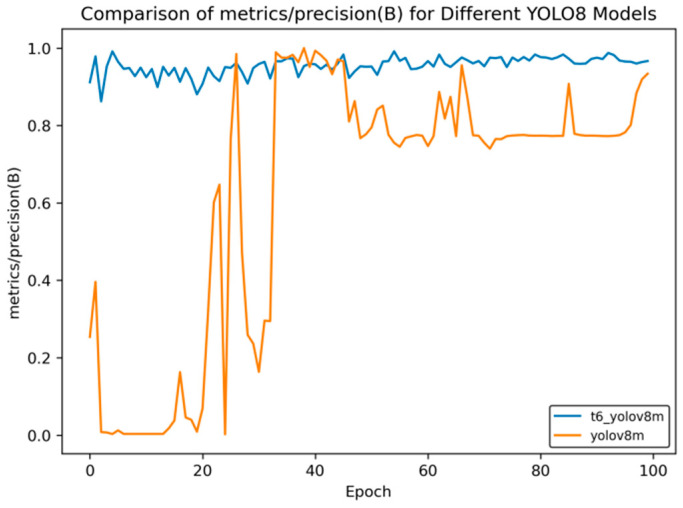
Comparison curve of accuracy after course learning strategies.

**Figure 16 sensors-25-02946-f016:**
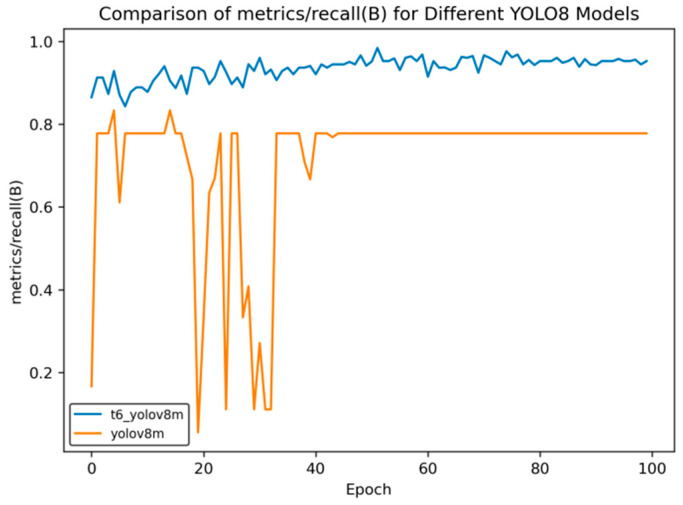
Comparison curve of recall rate after course learning strategy.

**Figure 17 sensors-25-02946-f017:**
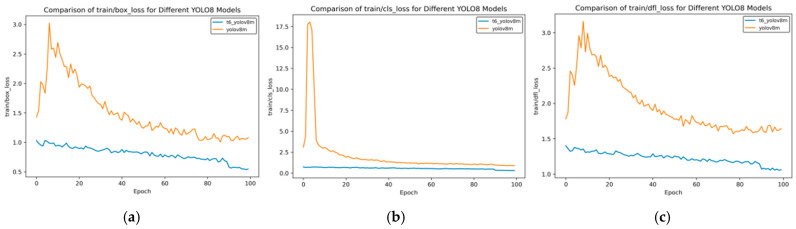
Train loss curve after course learning strategy. (**a**) Train box loss; (**b**) Train class loss; and (**c**) Train dfl loss.

**Figure 18 sensors-25-02946-f018:**
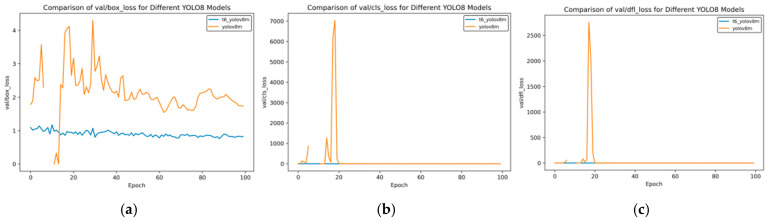
Val loss curve after course learning strategy. (**a**) Val box loss; (**b**) val class loss; and (**c**) val dfl loss.

Overall, the t6_yolov8m model trained through transfer learning significantly surpasses the YOLOv8m model in terms of both loss convergence and detection performance.

We used the finally obtained optimal model to conduct object detection on the data of the validation set of spark images from different angles, obtaining the following set of detection results.

Figure 19 shows the detection results of 45-degree spark images, Figure 20 shows the detection results of frontal spark images, and Figure 21 shows the detection results after fusing spark images from multiple angles. The detection results of side spark images are in the upper left corner, and the rest are the detection results of frontal spark images. The blue boxes represent the detected target areas, and the numbers on the boxes, such as “0.9” and “0.8”, represent the detection confidence. The closer the value is to 1, the higher the prediction credibility of the model for the presence of a target in this area. Judging from the detection results, the following inferences can be made:Confidence Level: The confidence of most detection results is 0.9, indicating that the model is quite certain about the detection of these sparks; only the detection confidence in one image is 0.8, which is relatively lower, but still has a high level of credibility.Detection Effect: For sparks in different angles and scenarios, the YOLOv8 model, after transfer learning, can identify and frame them well, indicating that this model has a certain degree of effectiveness and stability in the detection scenarios of industrial processing sparks of this kind.

**Figure 19 sensors-25-02946-f019:**
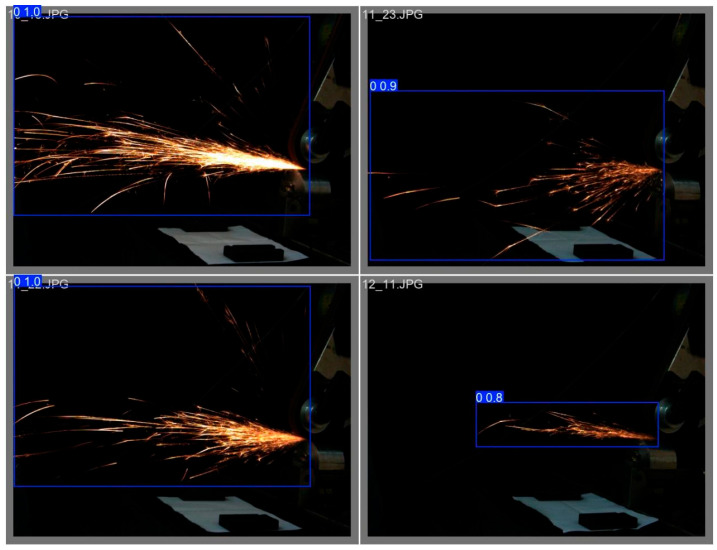
Target detection results of 45-degree spark images.

**Figure 20 sensors-25-02946-f020:**
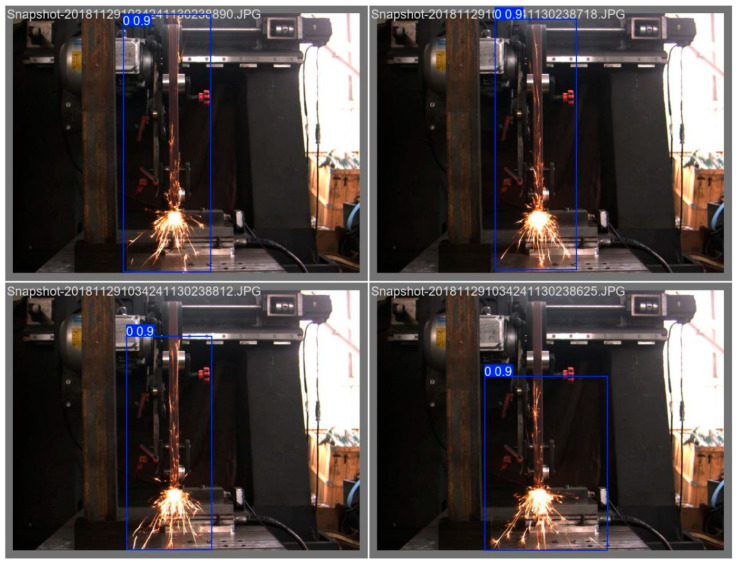
Target detection results of front spark images.

**Figure 21 sensors-25-02946-f021:**
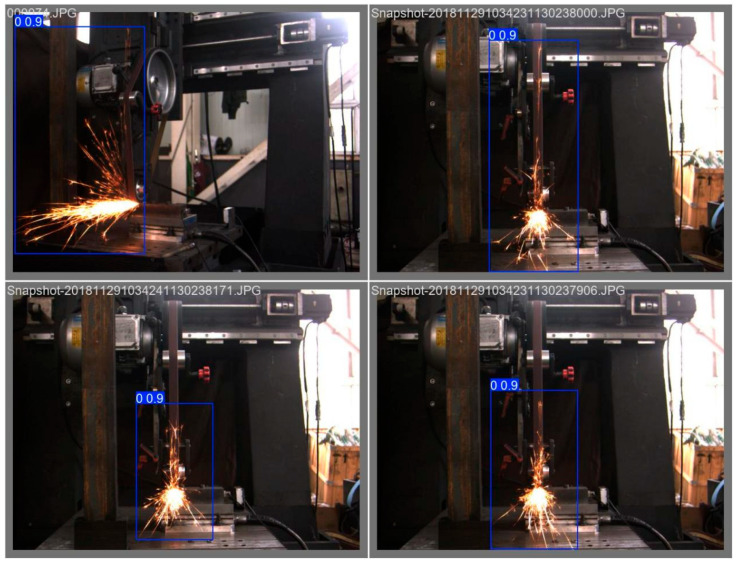
Object detection results of spark images with multiple data fusion.

During the testing process, we found that the object-detection performance for some spark images was sub-optimal, as shown in Figure 22, with a confidence level of only 0.62. In terms of the image itself and external interference: First, there are issues with image characteristics. The low contrast between the spark area and the background, along with uneven lighting, makes it difficult for the model to accurately extract spark features. When the brightness is similar, confusion is likely to occur. Second, there is external interference. Although there is no obvious occlusion in the image, other objects or splashes may interfere with the model’s recognition in the processing scenario.

From the perspectives of training data and the model itself, there is a deviation in the distribution of training data. If there are significant differences in aspects such as angles and lighting between the training set and the test images, the generalization ability of the model will be affected.

Figure 23 shows the precision–recall (PR) curve of the model. The horizontal axis represents the recall rate, which reflects the proportion of positive samples correctly detected by the model; the vertical axis represents the precision rate, which represents the proportion of truly positive samples among the detected positive samples. When the recall rate increases from 0 to nearly 1, the precision rate remains high, indicating that the detection accuracy is high and the number of false positives is low in the early stage. When the recall rate approaches 1, the precision rate drops sharply, suggesting that more false positives are introduced in an attempt to increase the recall rate.

The value “0 0.974”: The value “0” may refer to a specific class, corresponding to a mean average precision (mAP) value of 0.974. A higher mAP indicates better detection performance for this class. “all classes 0.974 mAP@0.5” means that the mean average precision of all classes is 0.974 at an Intersection over Union (IoU) threshold of 0.5, reflecting the comprehensive performance of the model. Overall, this YOLOv8 model combines both precision and recall capabilities, demonstrating excellent comprehensive performance.

In addition, to verify the performance of the modified Yolo v8 model, we also trained the Faster RCNN and Yolo v5 models on the dataset for 100 Epochs and obtained their respective evaluation metrics. These were then compared with those of the modified Yolo v8 model. The data obtained are shown in Table 2.

As can be seen from Table 2, the modified Yolo v8 model demonstrates the best performance in terms of precision, recall, and mean value, and has the lowest average value for various loss rates. This indicates that it has advantages in the accuracy of object recognition, the ability to identify positive-class objects, and the degree of model training fitting. It outperforms the Faster-RCNN and Yolo v5 models.

### 4.4. Experimental Results

Based on the comprehensive comparison and analysis above, the following results were obtained:(1)Comparison of Detection Accuracy

The experimental results show that, on the self-built multi-perspective dataset, the multi-perspective spark image detection method based on YOLOv8 transfer learning proposed in this paper achieves an average detection accuracy of 98.7%. Compared with the original YOLOv8 model, this comprises an increase of 14.2%, which fully demonstrates that a series of innovative methods adopted in this paper, such as transfer learning, dynamic anchor box optimization, attention-guided feature alignment module, and curriculum learning strategy, can effectively improve the detection accuracy of the model for multi-perspective spark images. It highlights the advantages of the method proposed in this paper in cross-perspective detection tasks. Through a detailed analysis of the detection accuracy under different perspectives, it was found that under the frontal perspective, the detection accuracy of each model is relatively high, but under the side and 45° angle perspectives, the advantages of the method in this paper are more obvious. This is because the dynamic anchor box optimization and the attention-guided feature alignment module can better adapt to the shape and feature differences of sparks under different perspectives, and the curriculum learning strategy helps the model to more effectively learn the key features required for cross-perspective detection, thus improving the detection performance of the model under complex perspectives.

(2)Analysis of Recall Rate and Precision Rate

In terms of the recall rate, the method in this paper reaches 96.5%, which is significantly higher than the 93.2% attained by the original YOLOv8 model. This indicates that the method in this paper can more comprehensively detect the spark targets in the images and reduce the occurrence of missed detections. In terms of the precision rate, that of the proposed method is 97.8%, which is also higher than that of other comparative models. A higher precision rate means that among the spark targets detected by the model, the proportion of truly correct ones is higher, effectively reducing the false-positive rate. Through the comprehensive analysis of the recall rate and the precision rate, it can be seen that the method in this paper achieves a good balance between the recall rate and the precision rate in the multi-perspective spark image detection task, and can accurately and comprehensively detect the spark targets under different perspectives, providing reliable data support for the subsequent monitoring and control of the grinding process.

(3)Inference Speed Test

On the NVIDIA RTX 4090, the inference speed of the method in this paper reaches 55 FPS, meeting the real-time requirements of industrial online monitoring and having better real-time performance. This benefits from the optimization strategy for the YOLOv8 model, which improves the detection accuracy of the model without significantly increasing the computational complexity, enabling it to quickly and accurately detect the belt grinding sparks in practical industrial applications.

(4)Verification of Model Generalization Ability

To further verify the generalization ability of the model, the trained model was applied to a new multi-perspective spark image test set that was not involved in the training. The test set includes spark images from different grinding working conditions and different workpiece materials. The experimental results show that the method proposed in this paper can still maintain a high detection accuracy on the new test set, with the average detection accuracy reaching 97.5%. However, the performance of other comparative models on the new test set decreased to varying degrees. In particular, the detection accuracy of the original YOLOv8 model and the YOLOv8 transfer learning model without the curriculum learning strategy decreased significantly. This proves that by adopting methods such as dynamic anchor box optimization, the attention-guided feature alignment module, and the curriculum learning strategy, the method proposed in this paper effectively improves the generalization ability of the model, enabling it to better adapt to the multi-perspective spark image detection tasks under different working conditions and having stronger practical application value.

## 5. Conclusions

Aiming at problems such as low accuracy, poor efficiency, and insufficient model generalization ability faced in the detection of belt grinding sparks during the manual grinding process of blades, this study proposes a multi-perspective spark image detection method based on YOLOv8 transfer learning. Through a series of innovative measures, including constructing a multi-pose spark image dataset, designing a cross-perspective transfer learning framework based on dynamic anchor box optimization, introducing an attention-guided feature alignment module, and adopting a curriculum learning strategy, the detection performance of the model has been significantly improved. The experimental results show that on the self-built multi-perspective dataset, after multiple transfer learning processes, an average detection accuracy of 98.7% is achieved, which is an increase of more than 15% compared with the original YOLOv8 model, and other evaluation indicators have also been significantly improved. The inference speed reaches 55 FPS on the NVIDIA RTX 4090, meeting the requirements of industrial online monitoring. The research results provide key technical support for the intelligent prediction of the material removal rate in the precision machining of blades and have the potential for rapid deployment in industrial scenarios.

This work has certain limitations. In terms of the dataset, although a multi-pose spark image dataset was constructed, the variations in blade materials and grinding process parameters (such as the grinding speed and pressure) in the actual industrial environment can lead to a greater diversity of spark images. The existing dataset cannot comprehensively cover these variations, which affects the application performance of the model in a wide range of scenarios. At the model level, while innovative measures have improved performance, they have also increased the complexity of the model. This results in a longer training time and makes it prone to facing the problem of insufficient computing resources when deploying the model on devices with limited resources. In the experimental comparison, the model is only compared with the original YOLOv8 model, lacking comparisons with other advanced object detection models based on Transformer and the like. As a result, it is difficult to comprehensively evaluate the advantages of the proposed method.

In addition, methods such as reinforcement learning and ensemble learning can also be adopted. Reinforcement learning enables the model to learn and make decisions in different environments and tasks. By using the detection results as feedback signals, it continuously adjusts the model’s parameters and strategies to optimize the detection process, improve the model’s performance and adaptability, and facilitate more efficient detection in practical applications. Ensemble learning combines multiple different models through methods such as Bagging and Boosting. For example, multiple YOLOv8 models can be trained, each on a different subset of the dataset, and then the prediction results of these models can be integrated to improve the stability and accuracy of the model and obtain more reliable detection results.

## Figures and Tables

**Figure 1 sensors-25-02946-f001:**
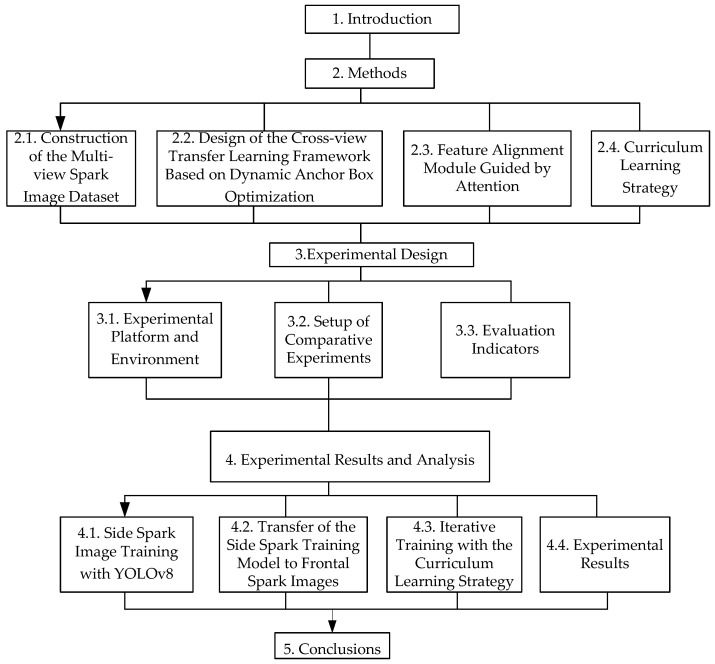
Diagram of the overall paper structure.

**Figure 2 sensors-25-02946-f002:**
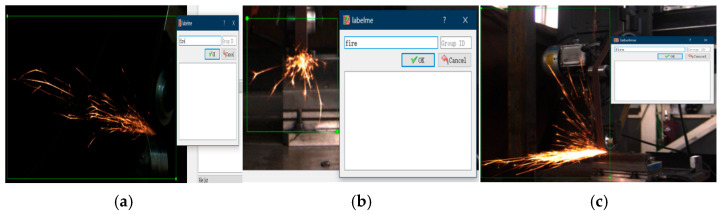
Spark image annotation maps. (**a**) Horizontal 45-degree view; (**b**) front view (front spark image); and (**c**) left view (side spark image).

**Figure 3 sensors-25-02946-f003:**
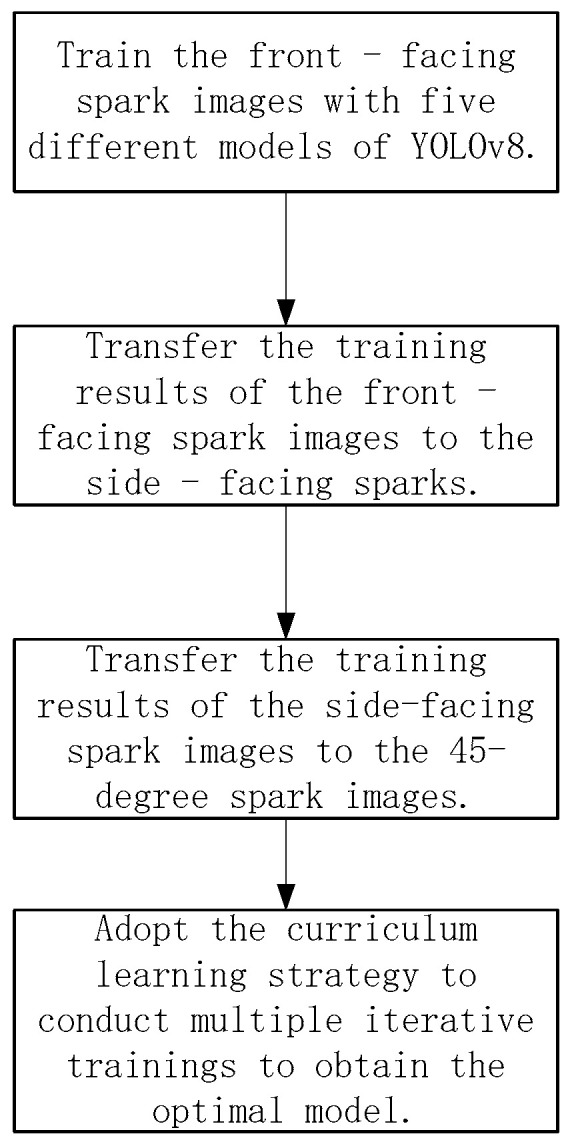
Transfer learning methods and strategies.

**Figure 4 sensors-25-02946-f004:**
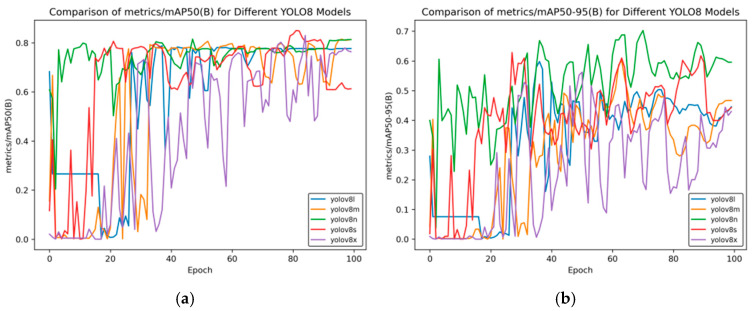
mAP curve of YOLOv8 training for side spark images. (**a**) mAP50 curve; (**b**) mAP50-95 curve.

**Figure 5 sensors-25-02946-f005:**
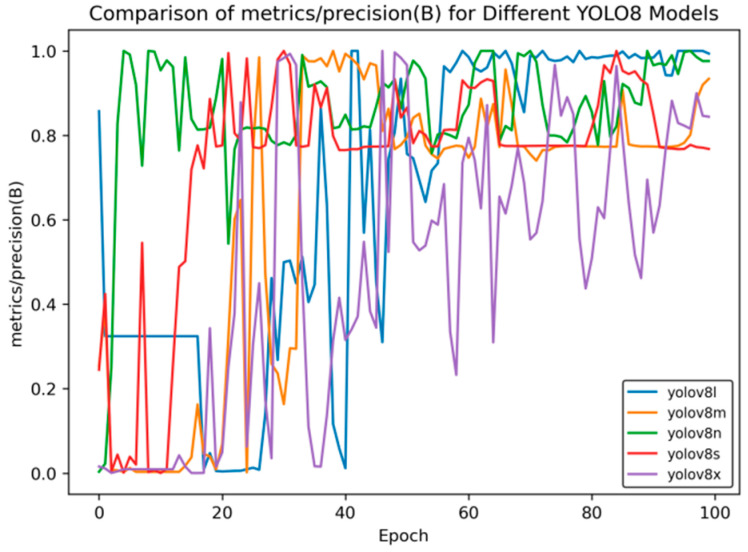
Precision rate curve of YOLOv8 training for side spark images.

**Figure 6 sensors-25-02946-f006:**
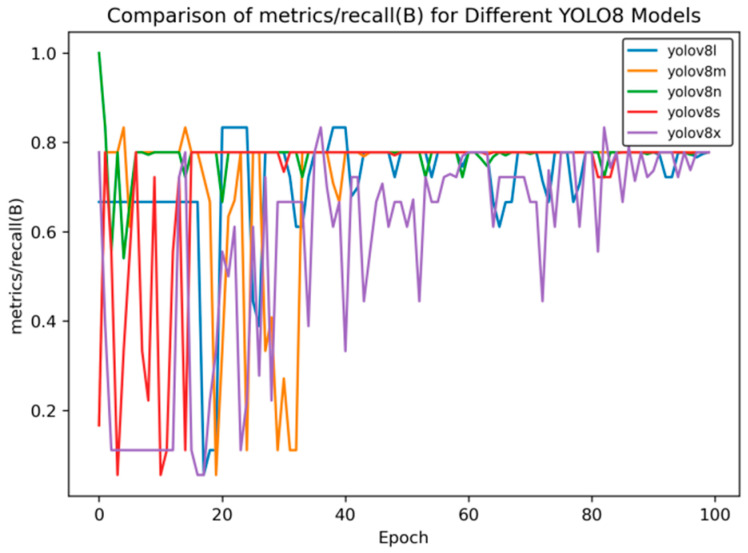
Recall rate curve of YOLOv8 training for side spark images.

**Figure 7 sensors-25-02946-f007:**
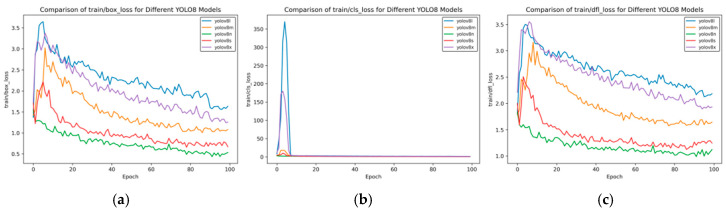
Train loss curve of YOLOv8 training for side spark images. (**a**) Train box loss; (**b**) train class loss; and (**c**) train dfl loss.

**Figure 9 sensors-25-02946-f009:**
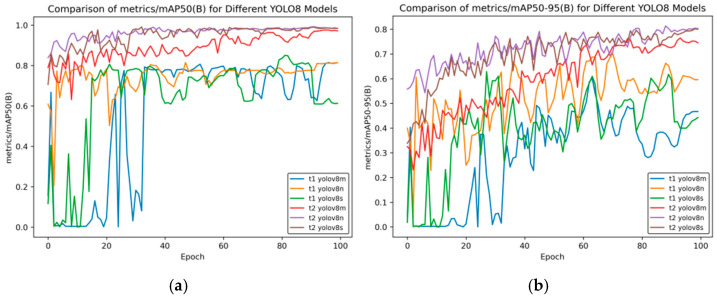
Comparison curve of mAP in transfer learning. (**a**) mAP50 curve; (**b**) mAP50-95 curve.

**Figure 10 sensors-25-02946-f010:**
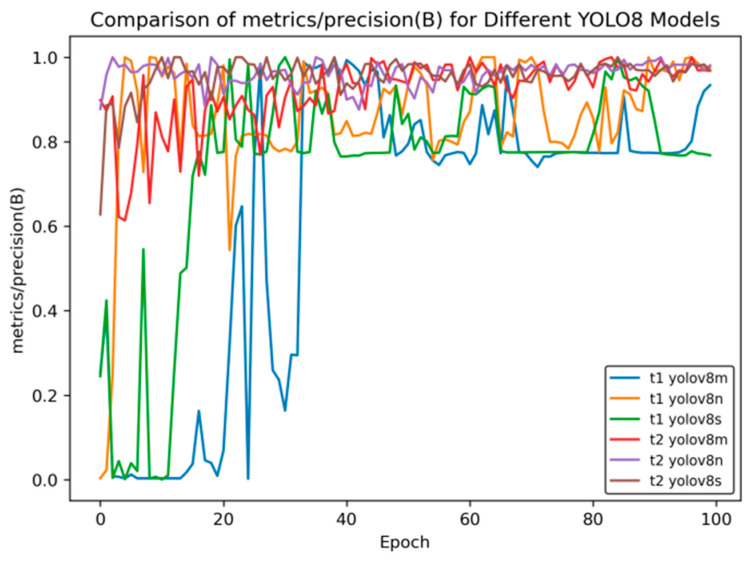
Comparison curve of precision after transfer learning.

**Figure 11 sensors-25-02946-f011:**
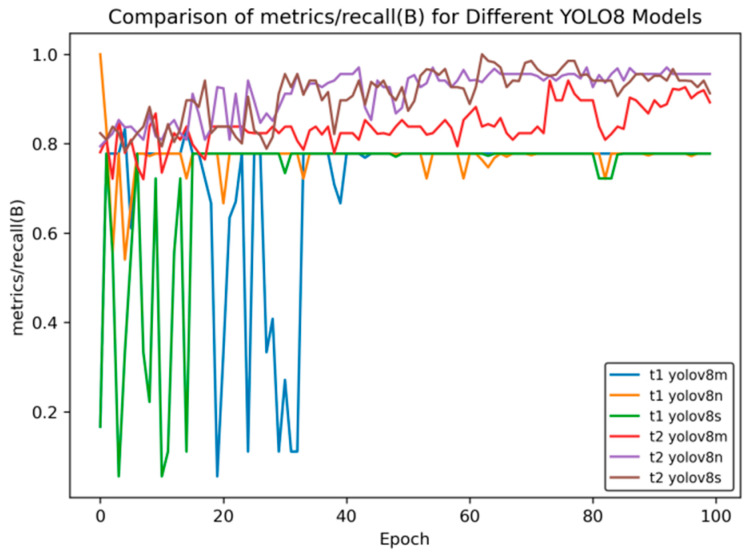
Comparison curve of recall after transfer learning.

**Figure 12 sensors-25-02946-f012:**
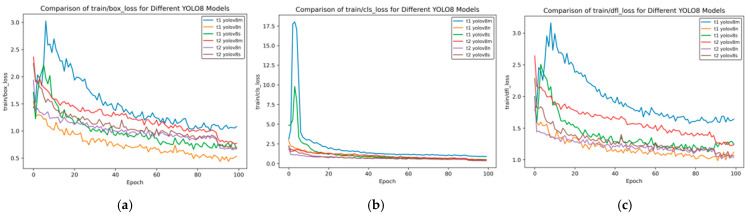
Train loss curve of transfer training. (**a**) Train box loss; (**b**) train class loss; and (**c**) train dfl loss.

**Figure 22 sensors-25-02946-f022:**
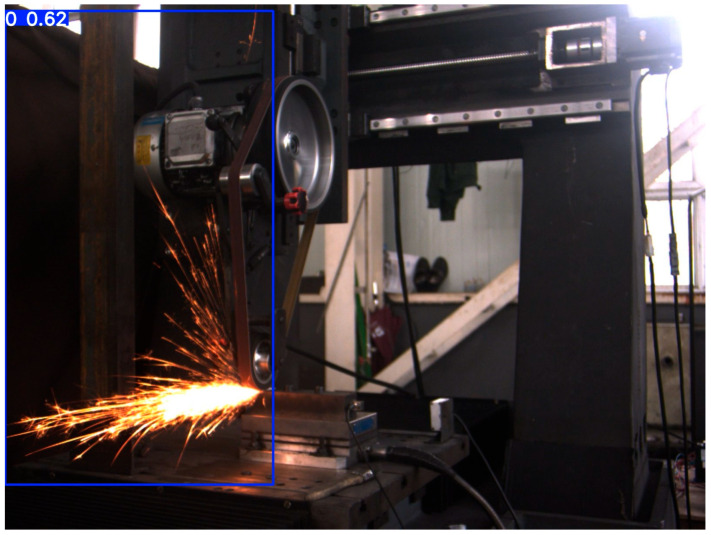
Object detection results of spark images.

**Figure 23 sensors-25-02946-f023:**
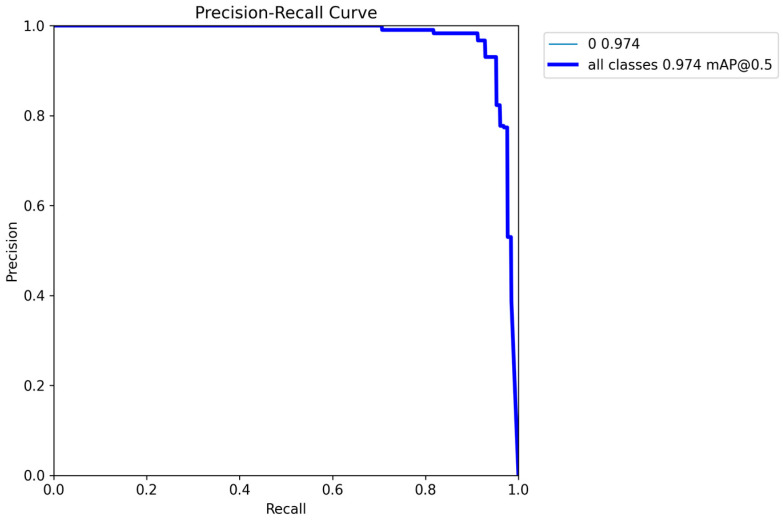
PR curve.

**Table 1 sensors-25-02946-t001:** Sand belt grinding parameter settings.

Number	Belt Speed (m/s)	Workpiece Feed Rate (mm/s)	Theoretical Grinding Depth (mm)
1	26	1.5	0.3
2	28	2	0.3
3	30	2.5	0.3
4	32	3	0.3
5	34	3.5	0.3
6	36	4	0.3
7	38	4.5	0.3

**Table 2 sensors-25-02946-t002:** Comparison of Model Performance.

Network Model	Precision	Total Loss	Recall
Faster RCNN	91.8	0.031	90.5
Yolov5	95.1	0.025	91.3
Modified Yolo v8	98.7	0.012	93.5

## Data Availability

The data presented in this study are available on request from the corresponding author.

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
