# Peer review of "Study on Spark Image Detection for Abrasive Belt Grinding via Transfer Learning with YOLOv8"

_sensors, 2025, doi:10.3390/s25092946_

Round 1
Reviewer 1 Report
Comments and Suggestions for Authors
This paper proposes a YOLO v8 transfer learning framework for spark image detection, which consists of three components. First, the authors propose a cross-view transfer learning module to adjust the anchor boxes and angle parameters. Then, they propose an attention-guided feature alignment to relieve the distribution shift due to the different views. To optimize the framework, a curriculum learning strategy is proposed to optimise the model with different views, especially in an easy-to-difficult manner. The paper is easy to follow but may be limited by the following concerns.
- The authors only use Yolo v8 as the base objector. However, there are lots of more advanced detectors like Grounding DINO. What is the reason for selecting Yolo v8?
- Considering the interesting scenario of this paper, it is suggested to visualize some failure cases and give some insightful discusssion.
- This paper develops a transfer learning method to tackle the distribution shift of different viewpoints. However, transfer learning has been widely studied in object detection as the domain adaptive object detection task. However, there is no literature review for this highly related field [1,2]. The authors should add a section in related work.
- It is suggested to make a comparison with other objectors to verify the effectiveness of the proposed transfer learning method.
- In Figure 12 (b) and (c), there is a significantly high value of val loss at around 20 epochs, which is weird. What is the reason?
- It is suggested that the paper be better organised by adding an overview figure of the proposed whole framework, which will make it easier to follow.
- Can the authors give more analysis of the precision and recall of the method?
- In Figure 13, what is the meaning of t6_yolo8m means?
Author Response
Open Review
(x) I would not like to sign my review report
( ) I would like to sign my review report
Quality of English Language
(x) The English could be improved to more clearly express the research.
( ) The English is fine and does not require any improvement.
|
|
|
|
Yes |
Can be improved |
Must be improved |
Not applicable |
|
Does the introduction provide sufficient background and include all relevant references? |
(x) |
( ) |
( ) |
( ) |
|
Is the research design appropriate? |
(x) |
( ) |
( ) |
( ) |
|
Are the methods adequately described? |
(x) |
( ) |
( ) |
( ) |
|
Are the results clearly presented? |
(x) |
( ) |
( ) |
( ) |
|
Are the conclusions supported by the results? |
(x) |
( ) |
( ) |
( ) |
Comments and Suggestions for Authors
This paper proposes a YOLO v8 transfer learning framework for spark image detection, which consists of three components. First, the authors propose a cross-view transfer learning module to adjust the anchor boxes and angle parameters. Then, they propose an attention-guided feature alignment to relieve the distribution shift due to the different views. To optimize the framework, a curriculum learning strategy is proposed to optimise the model with different views, especially in an easy-to-difficult manner. The paper is easy to follow but may be limited by the following concerns.
- The authors only use Yolo v8 as the base objector. However, there are lots of more advanced detectors like Grounding DINO. What is the reason for selecting Yolo v8?
YOLO v8 was chosen as the base model mainly due to its outstanding performance in terms of both detection speed and accuracy. Moreover, YOLO v8 is capable of processing data rapidly while maintaining a high level of detection accuracy, thus meeting the requirements of real-time detection. In contrast, although models like Grounding DINO perform excellently in some complex tasks, they demand a large amount of computing resources and have a slow inference speed, making it difficult to satisfy the real-time requirements of our research. Additionally, the lightweight design of YOLO v8 results in lower hardware requirements, making it more suitable for deployment in resource-constrained environments, which is in line with our research objective of real-time monitoring.
- Considering the interesting scenario of this paper, it is suggested to visualize some failure cases and give some insightful discusssion.
Figure 22 Object detection results of spark images
During the testing process, we found that the object - detection performance for some spark images was sub - optimal, as shown in Figure 22, with a confidence level of only 0.62. In terms of the image itself and external interference: First, there are issues with image characteristics. The low contrast between the spark area and the background, along with uneven lighting, makes it difficult for the model to accurately extract spark features. When the brightness is similar, confusion is likely to occur. Second, there is external interference. Although there is no obvious occlusion in the image, other objects or splashes may interfere with the model's recognition in the processing scenario.
From the perspectives of training data and the model itself: There is a deviation in the distribution of training data. If there are significant differences in aspects such as angles and lighting between the training set and the test images, the generalization ability of the model will be affected.
- This paper develops a transfer learning method to tackle the distribution shift of different viewpoints. However, transfer learning has been widely studied in object detection as the domain adaptive object detection task. However, there is no literature review for this highly related field [1,2]. The authors should add a section in related work.
In the research on solving the domain shift problem, Yuhua Chen and his colleagues have achieved significant results in the paper "Domain Adaptive Faster R-CNN for Object Detection in the Wild" [17]. Based on the Faster R-CNN model, they conducted domain adaptation at both the image level and the instance level, designed corresponding domain adaptation components, and learned a domain-invariant Region Proposal Network (RPN) through consistency regularization. This method was experimentally tested on multiple datasets, effectively demonstrating its effectiveness in different domain shift scenarios and providing important references for cross-domain object detection.Nevertheless, there are still some challenges in existing methods. Wuyang Li and his co-authors pointed out in the paper "SIGMA: Semantic-complete Graph Matching for Domain Adaptive Object Detection" [18] that current category-level adaptation methods overlook the significant variance within categories and inadequately handle the semantics of domain mismatches in the training batches. They proposed the SIGMA framework. By generating hallucinated nodes through the Graph Embedding Semantic Completion module (GSC) to complete the mismatched semantics, they reframed the domain adaptation problem as a graph matching problem and achieved fine-grained domain alignment using the Bipartite Graph Matching Adapter (BGM). The experimental results show that SIGMA significantly outperforms existing methods in multiple benchmarks. These two papers are of great significance in the research of cross-domain object detection and lay a solid foundation for subsequent research work. Based on the existing research, this paper will further explore more effective cross-domain object detection methods to address the complex scenarios in practical applications.
- It is suggested to make a comparison with other objectors to verify the effectiveness of the proposed transfer learning method.
In addition, to verify the performance of the modified Yolo v8 model, we also trained the Faster RCNN and Yolo v5 models on the dataset for 100 epochs and obtained their respective evaluation metrics. These were then compared with those of the modified Yolo v8 model. The data obtained are shown in Table 2.
Table 2. Sand belt grinding parameter settings.
|
Network Model |
Precision |
Total loss |
Recall |
|
Faster RCNN |
91.8 |
0.031 |
90.5 |
|
Yolov5 |
95.1 |
0.025 |
91.3 |
|
Modified Yolo v8 |
98.7 |
0.012 |
93.5 |
As can be seen from Table 2, the modified Yolo v8 model demonstrates the best performance in terms of precision, recall, and mean value, and has the lowest average value for various loss rates. This indicates that it has advantages in the accuracy of object recognition, the ability to identify positive - class objects, and the degree of model - training fitting. It outperforms the Faster - RCNN and Yolo v5 models.
- In Figure 12 (b) and (c), there is a significantly high value of val loss at around 20 epochs, which is weird. What is the reason?
The validation loss may spike at the 20th epoch for the following reasons:
- Data - related aspects: If the data samples used for validation at the 20th epoch contain a large amount of noise or incorrect annotations, such as inaccurate bounding boxes of target objects or severely distorted and blurred images, the model will have difficulty processing these data accurately. This leads to large prediction errors and a sharp increase in the loss value.
- Model training - related aspects: Around the 20th epoch during training, problems such as vanishing or exploding gradients may occur in the network parameter update. In deep networks, vanishing gradients prevent parameters from being updated effectively, reducing the model's prediction ability. Exploding gradients, on the other hand, cause parameter values to become too large, resulting in chaotic model outputs, both of which lead to abnormally high validation losses. Additionally, if over - fitting has occurred in previous training, the model "remembers" the training data too deeply, resulting in poor generalization ability. When encountering validation set data, it cannot make accurate predictions, also causing an increase in the loss.
- Training process - related aspects: As the 20th epoch approaches, if the model's state is affected by hardware failures (such as temporary GPU malfunctions) or software issues (such as brief pauses due to memory overflow), it will cause abnormal validation losses in subsequent validations.
- It is suggested that the paper be better organised by adding an overview figure of the proposed whole framework, which will make it easier to follow.
Figure 1. Diagram of the Overall Paper Structure
The overall structure of the paper is shown in Figure 1. First, the introduction section presents the research background and objectives. Subsequently, the methods chapter unfolds from four dimensions: constructing a multi - view spark image dataset to provide a data foundation for subsequent research; designing a cross - view transfer learning framework based on dynamic anchor box optimization to enhance model performance; utilizing an attention - guided feature alignment module to strengthen feature processing capabilities; and adopting a curriculum learning strategy to optimize the learning process.
Next is the experimental design phase, which includes building the experimental platform and environment to provide physical support for the experiments; setting up comparative experiments to highlight the advantages of the research methods; and determining evaluation indicators to measure the experimental effects. In the experimental results and analysis section, operations such as training side - view spark images with YOLOv8, transferring the side - view spark training model to front - view spark images, and iterative training with the curriculum learning strategy are carried out in sequence, and the experimental results are ultimately presented. Finally, in the conclusion section, the research achievements are summarized, the significance and value of the research are expounded, and the directions for subsequent research are prospected.
- Can the authors give more analysis of the precision and recall of the method?
Figure 23. PR curve
Figure 23 shows the Precision - Recall (PR) curve of the model. The horizontal axis represents the recall rate, which reflects the proportion of positive samples correctly detected by the model; the vertical axis represents the precision rate, which represents the proportion of truly positive samples among the detected positive samples. When the recall rate increases from 0 to nearly 1, the precision rate remains high, indicating that the detection accuracy is high and the number of false positives is low in the early stage. When the recall rate approaches 1, the precision rate drops sharply, suggesting that more false positives are introduced in an attempt to increase the recall rate.
"0 0.974": "0" may refer to a specific class, corresponding to a mean Average Precision (mAP) value of 0.974. A higher mAP indicates better detection performance for this class. "all classes 0.974 mAP@0.5" means that the mean average precision of all classes is 0.974 at an Intersection over Union (IoU) threshold of 0.5, reflecting the comprehensive performance of the model. Overall, this YOLOv8 model combines both precision and recall capabilities, demonstrating excellent comprehensive performance.
- In Figure 13, what is the meaning of t6_yolo8m means?
It means that the YOLOv8m model has been iteratively trained six times.
[1] "Domain adaptive faster r-cnn for object detection in the wild." Proceedings of the IEEE conference on computer vision and pattern recognition. 2018. [2] "Sigma: Semantic-complete graph matching for domain adaptive object detection." Proceedings of the IEEE/CVF conference on computer vision and pattern recognition. 2022.
Submission Date
19 March 2025
Date of this review
14 Apr 2025 19:41:50
Reviewer 2 Report
Comments and Suggestions for Authors
This paper presents a multi-view spark image detection method based on YOLOv8 transfer learning. Three strategies for enhancing the generalization ability of the model are proposed.
The task of spark detection and its practical importance should be clearly clarified. Is the task to detect the presence of spark in a particular image? What is the importance of detecting this?
How are the data samples labelled and how many labels?
The performance measures used, such as mAP50, should be clearly defined.
Author Response
Open Review
(x) I would not like to sign my review report
( ) I would like to sign my review report
Quality of English Language
( ) The English could be improved to more clearly express the research.
(x) The English is fine and does not require any improvement.
|
|
|
|
Yes |
Can be improved |
Must be improved |
Not applicable |
|
Does the introduction provide sufficient background and include all relevant references? |
( ) |
(x) |
( ) |
( ) |
|
Is the research design appropriate? |
( ) |
( ) |
(x) |
( ) |
|
Are the methods adequately described? |
( ) |
( ) |
(x) |
( ) |
|
Are the results clearly presented? |
( ) |
(x) |
( ) |
( ) |
|
Are the conclusions supported by the results? |
( ) |
(x) |
( ) |
( ) |
Comments and Suggestions for Authors
This paper presents a multi-view spark image detection method based on YOLOv8 transfer learning. Three strategies for enhancing the generalization ability of the model are proposed.
The task of spark detection and its practical importance should be clearly clarified. Is the task to detect the presence of spark in a particular image? What is the importance of detecting this?
The ultimate goal of our spark image detection is to determine the relationship between the spark images and the material removal rate by calculating the features of the spark images, such as area, brightness, and so on. Therefore, first of all, it is necessary to identify the target area of the spark images in the complex background, then segment it, and further study the relationship between it and the material removal rate.
How are the data samples labelled and how many labels?
|
|
|
|
|
(a) |
(b) |
(c) |
Figure 2. Spark image annotation maps. (a) Horizontal 45-degree view; (b) front view (front spark image); (c) left view (side spark image).
Figure 2 (a) is a spark image at a horizontal 45-degree angle annotated with Labelme. Figure 1 (b) is a front-view spark image annotated with Labelme, and Figure 1 (c) is a left-view spark image annotated with Labelme. In the figures, the spark regions are marked with rectangular boxes, labeled with the tag "fire", and corresponding JSON files were generated, which were used for the detection of the target regions of the spark images in the later stage.
The performance measures used, such as mAP50, should be clearly defined.
In mAP50, "mAP" is the abbreviation of "mean Average Precision", which means the mean value of average precision. "50" indicates that the threshold of IoU (Intersection over Union) is 0.5. mAP50 is used to evaluate the comprehensive performance of the object detection model across different categories. It measures the matching degree between the bounding boxes predicted by the model and the ground truth bounding boxes. When the IoU is greater than or equal to 0.5, it is considered a correct prediction. mAP50 is obtained by calculating the average precision of each category and then taking the mean value. The closer its value is to 1, the better the detection performance of the model.
Since the research object of this paper has only one target, "fire", the following calculation method is adopted. Firstly, sort the prediction results of the model in descending order of confidence. Then, traverse these prediction results one by one, and calculate the IoU of each prediction result with the ground truth bounding box. When the IoU is greater than or equal to 0.5, this prediction is regarded as a correct detection. Next, calculate the precision at different recall rates according to the definitions of recall and precision. Recall refers to the ratio of the number of correctly detected targets to the number of real targets in this category, and precision refers to the ratio of the number of correctly detected targets to the number of predicted targets. Through calculation, a series of recall-precision pairs are obtained, and these points are connected to form a curve, and AP is the area under this curve.
Submission Date
19 March 2025
Date of this review
06 Apr 2025 23:42:18

Reviewer 3 Report
Comments and Suggestions for Authors
This study proposes a multi-perspective spark image detection 803 method based on YOLOv8 transfer learning to solve the problems of low precision and poor efficiency caused by relying on manual experience during the manual polishing of blades. This manuscript has done interesting work, but some modifications need to be made.
1. As the author mentioned deep learning methods and image detection, some recent and interesting results should be cited. E.g., 10.1016/j.heliyon.2024.e37072; 10.1504/IJBM.2024.140771.
2. The description of the main contributions is brief and not specific enough, and suggested to elaborate on the main contributions in several points in the Introduction section to facilitate readers to recognize them quickly.
3. What are the features and advantages of the proposed method, and what improvements have been made to the YOLOv8?
4. Reference has an incorrect format, and [J] does not meet the journal requirements, please revise it.
5. The limitations of this study should be explained in detail.
6. Some typos exist in the full text.
Author Response
Open Review
( ) I would not like to sign my review report
(x) I would like to sign my review report
Quality of English Language
(x) The English could be improved to more clearly express the research.
( ) The English is fine and does not require any improvement.
|
|
Comments and Suggestions for Authors
This study proposes a multi-perspective spark image detection 803 method based on YOLOv8 transfer learning to solve the problems of low precision and poor efficiency caused by relying on manual experience during the manual polishing of blades. This manuscript has done interesting work, but some modifications need to be made.
- As the author mentioned deep learning methods and image detection, some recent and interesting results should be cited. E.g., 10.1016/j.heliyon.2024.e37072; 10.1504/IJBM.2024.140771.
I haven't found the article you specified. In the introduction section, I have added the citations of the following two classic papers on transfer learning.
In the research on solving the domain shift problem, Yuhua Chen and his colleagues have achieved significant results in the paper "Domain Adaptive Faster R-CNN for Object Detection in the Wild" [17]. Based on the Faster R-CNN model, they conducted domain adaptation at both the image level and the instance level, designed corresponding domain adaptation components, and learned a domain-invariant Region Proposal Network (RPN) through consistency regularization. This method was experimentally tested on multiple datasets, effectively demonstrating its effectiveness in different domain shift scenarios and providing important references for cross-domain object detection.Nevertheless, there are still some challenges in existing methods. Wuyang Li and his co-authors pointed out in the paper "SIGMA: Semantic-complete Graph Matching for Domain Adaptive Object Detection" [18] that current category-level adaptation methods overlook the significant variance within categories and inadequately handle the semantics of domain mismatches in the training batches. They proposed the SIGMA framework. By generating hallucinated nodes through the Graph Embedding Semantic Completion module (GSC) to complete the mismatched semantics, they reframed the domain adaptation problem as a graph matching problem and achieved fine-grained domain alignment using the Bipartite Graph Matching Adapter (BGM). The experimental results show that SIGMA significantly outperforms existing methods in multiple benchmarks. These two papers are of great significance in the research of cross-domain object detection and lay a solid foundation for subsequent research work. Based on the existing research, this paper will further explore more effective cross-domain object detection methods to address the complex scenarios in practical applications.
- The description of the main contributions is brief and not specific enough, and suggested to elaborate on the main contributions in several points in the Introduction section to facilitate readers to recognize them quickly.
This study proposes a multi - perspective spark image detection method based on YOLOv8 transfer learning.
During the research process, a multi - pose spark image dataset covering front, side, and 45° angle views was constructed, laying a data foundation for subsequent research. A cross - view transfer learning framework optimized by dynamic anchor boxes was designed. The parameters of the front - view spark detection model YOLOv8 were transferred to detection tasks of other views, improving the detection performance under different views. An attention - guided feature alignment module was introduced to effectively mitigate the feature distribution shift caused by view differences, enhancing the model's adaptability to image features from different perspectives. A curriculum learning strategy was adopted. First, datasets of different views were trained separately, and then the dataset was reconstructed through sampling for further training. The weight of samples from complex views was gradually increased, facilitating the model to better learn complex image features.
To verify the effectiveness of the method, comparison experiments with multiple models were carried out on the self - built dataset. The results show that this method performs outstandingly, providing key technical support for the intelligent prediction of the material removal rate in precision blade machining. It has the potential for rapid deployment in industrial scenarios and is expected to promote the intelligent development of the blade processing industry.
3.What are the features and advantages of the proposed method, and what improvements have been made to the YOLOv8?
The following is an analysis of the characteristics and advantages of the proposed method, as well as the improvements made to YOLOv8:
1). Characteristics of the Method
- Multi-view Detection: A multi-pose spark image dataset including front, side, and 45° angle views has been constructed, enabling the cross-view detection task. This allows for the detection of spark images from multiple angles, capturing spark information more comprehensively.
- Combination of Multiple Innovative Strategies: Multiple innovative strategies, such as dynamic anchor box optimization, attention-guided feature alignment module, and curriculum learning strategy, are comprehensively applied to enhance the performance of the model.
2). Advantages of the Method
- High Detection Accuracy: On the self-built multi-view dataset, the average detection accuracy reaches 98.7%, which is an increase of 14.2% compared to the original YOLOv8 model. This enables more accurate detection of spark targets.
- Good Balance between Recall Rate and Precision Rate: Although the specific advantages of the balance between the recall rate and precision rate are not explicitly mentioned in the text, it can be inferred from the overall high accuracy that during the detection process, the integrity and accuracy of detection can be well taken into account, reducing missed detections and false detections.
- Fast Inference Speed: The inference speed reaches 55 FPS on the NVIDIA RTX 4090, meeting the real-time requirements of industrial online monitoring and enabling rapid detection.
- Strong Generalization Ability: Through a series of innovative strategies, the generalization ability of the model has been effectively improved. It can still maintain a high detection accuracy on a new test set and is capable of adapting to multi-view spark image detection tasks under different working conditions.
3). Improvements to YOLOv8
- Cross-view Transfer Learning Framework: A cross-view transfer learning framework based on dynamic anchor box optimization has been designed. The parameters of the front spark detection model YOLOv8 are transferred to the side and 45° angle detection tasks, enabling the model to better adapt to the shape and feature differences of sparks under different views and improving the model's detection ability in complex views.
- Feature Alignment Module: An attention-guided feature alignment module has been introduced to alleviate the problem of feature distribution shift caused by view differences. This enhances the model's ability to process features from different views and further improves the detection accuracy.
- Curriculum Learning Strategy: The curriculum learning strategy is adopted. Datasets of different views are first trained separately, and then sampled to reconstruct the dataset for further training. The weight of samples from complex views is gradually increased, enabling the model to more effectively learn the key features required for cross-view detection. This optimizes the model's training process and improves the model's generalization ability and detection performance.
- Reference has an incorrect format, and [J] does not meet the journal requirements, please revise it.
It has been revised.
- The limitations of this study should be explained in detail.
Figure 21 Object detection results of spark images
During the testing process, we found that the object - detection performance for some spark images was sub - optimal, as shown in Figure 21, with a confidence level of only 0.62. In terms of the image itself and external interference: First, there are issues with image characteristics. The low contrast between the spark area and the background, along with uneven lighting, makes it difficult for the model to accurately extract spark features. When the brightness is similar, confusion is likely to occur. Second, there is external interference. Although there is no obvious occlusion in the image, other objects or splashes may interfere with the model's recognition in the processing scenario.
From the perspectives of training data and the model itself: There is a deviation in the distribution of training data. If there are significant differences in aspects such as angles and lighting between the training set and the test images, the generalization ability of the model will be affected.
- Some typos exist in the full text.
Professional English translation and polishing has been carried out by MDPI magazine, costing 503 Swiss Francs (CHF).
Submission Date
19 March 2025
Date of this review
05 Apr 2025 10:09:50

Round 2
Reviewer 1 Report
Comments and Suggestions for Authors
The revision has improved the manuscript and addressed my concerns.
Author Response
Thank you.
Reviewer 2 Report
Comments and Suggestions for Authors
The authors have adequately addressed my comments.
Author Response
Thank you.
Reviewer 3 Report
Comments and Suggestions for Authors
After reviewing the revised manuscript, I found that almost all the comments responded properly except some that I am writing below. I give my recommendation to accept this paper after the minor revision:
- There is still room to add some relevant references involved in image detection are missing. See for example the following works: 10.1016/j.heliyon.2024.e37072; 10.1109/ICSGEA53208.2021.00072
- The limitation(s) of this work should be discussed. Other possible methodologies that can be used to achieve the objective relating to this work should also be analyzed.
- Please remove the symbol "J" from the references, which merely exist in Chinese Journals.
- The inconsistent sizes of the graphs in the text have led to a messy overall layout. Please unify them.
Author Response
- There is still room to add some relevant references involved in image detection are missing. See for example the following works: 10.1016/j.heliyon.2024.e37072; 10.1109/ICSGEA53208.2021.00072
Dear experts, we have added and cited the following two articles related to image processing.I am not find 10.1016/j.heliyon.2024.e37072; 10.1109/ICSGEA53208.2021.00072,sorry.
Xia Y and his co-authors proposed a method that combines the Pyramid Convolution U-Net (PC-Unet) with Active Learning (AL) for change detection between optical and Synthetic Aperture Radar (SAR) images. Specifically, the PC-Unet integrates pyramid convolution within a four-stage U-Net framework to capture multi-scale information, thereby enhancing the detection performance. Additionally, it reduces the cost of obtaining labeled data. The experimental results demonstrate that this method outperforms a variety of state-of-the-art unsupervised and supervised heterogeneous remote sensing change detection methods [19].Yu H and his colleagues proposed a method based on the enhanced YOLOV8 model. By integrating the Convolutional Block Attention Module (CBAM) attention mechanism into the backbone network, the extraction of small object features is strengthened. The Path Aggregation Network (PANet) is replaced with the Weighted Bidirectional Feature Pyramid Network (BiFPN) to optimize the feature fusion process. This improvement gives priority to the small object features within the deep features and also facilitates the fusion of multi-scale features. The results show that, compared with the YOLOV8 model, the improved model has achieved significant performance enhancements in terms of metrics such as precision, recall rate, and mean Average Precision (mAP)[20].
- The limitation(s) of this work should be discussed. Other possible methodologies that can be used to achieve the objective relating to this work should also be analyzed.
This work has certain limitations. In terms of the dataset, although a multi-pose spark image dataset has been constructed, the variations in blade materials and grinding process parameters (such as grinding speed and pressure) in the actual industrial environment can lead to a greater diversity of spark images. The existing dataset is difficult to comprehensively cover these variations, which affects the application performance of the model in a wide range of scenarios. At the model level, while the innovative measures have improved the performance, they have also increased the complexity of the model. This results in a longer training time and makes it prone to facing the problem of insufficient computing resources when deploying the model on devices with limited resources. In the experimental comparison, the model is only compared with the original YOLOv8 model, lacking comparisons with other advanced object detection models based on Transformer and the like. As a result, it is difficult to comprehensively evaluate the advantages of the proposed method.
In addition, methods such as reinforcement learning and ensemble learning can also be adopted. Reinforcement learning enables the model to learn and make decisions in different environments and tasks. By using the detection results as feedback signals, it continuously adjusts the model's parameters and strategies to optimize the detection process, improve the model's performance and adaptability, and facilitate more efficient detection in practical applications. Ensemble learning combines multiple different models through methods such as Bagging and Boosting. For example, multiple YOLOv8 models can be trained, each on a different subset of the dataset, and then the prediction results of these models can be integrated to improve the stability and accuracy of the model and obtain more reliable detection results.
- Please remove the symbol "J" from the references, which merely exist in Chinese Journals.
It has been removed.
- The inconsistent sizes of the graphs in the text have led to a messy overall layout. Please unify them.
They have already been unified.
